# Longitudinal protein profiling of blood during childhood into early adulthood

Sofia Bergström [1,7], Sophia Björkander [2,7] ✉, María Bueno Álvez [1], Simon Kebede Merid[2], Hanna Danielsson [3,4], Anna Bergström[5,6], Inger Kull [2,4], Anne-Sophie Merritt[5,6], Fredrik Edfors [1], Susanna Klevebro[2,4], Mathias Uhlén [1], Peter Nilsson [1,8] & Erik Melén [2,4,8]

Proteomic research enhances our understanding of health- and disease-related biological processes. Protein profiling during healthy childhood provides important insights into normal physiological development. We longitudinally measured 5416 plasma proteins at four follow-ups during childhood (4-, 8-, 16 years) and early adulthood (24 years) in 100 randomly selected subjects participating in a population-based Swedish cohort, using Olink Explore HT. In total, 3509 proteins were included in the analysis. 54% of the proteins were found to be associated with age, and we observed several protein trajectories from childhood to adulthood based on clustering. In addition to proteins involved in bone, teeth and cartilage formation, we identified differences in proteins involved in neural function, drug metabolism, and hormonal control. There were pronounced sex-related differences in protein levels, particularly at follow-ups 16 and 24, characterized by, for example, growth, response to stimuli and regulation of catabolic processes. We demonstrate dynamic age- and sex-related changes in protein levels during the first two decades of life. Our study results may serve as an important resource in understanding human physiological development, disease etiology, and for future protein biomarker research.

The human circulating proteome provides a window and access to cellular- and tissue phenotypes and functions throughout the body in both homeostasis and disease. In the era of precision medicine, proteomic research has the potential to increase our understanding of not only disease etiology and pathogenesis in terms of risk factors, pathomechanisms and disease prediction, but also in improving diagnosis and treatment in terms of disease detection, distinction of disease phenotypes and endotypes, prediction of treatment responsiveness and clinical decision making[1]. Multiple cross-sectional studies have shown the potential usability of circulating proteins as biomarkers of diseases, and proteomic aging clocks reveal non-linear alterations in the human proteome with distinct waves of proteome changes and associations with age, sex, health, disease and mortality[2–10].

While one-time measurements of proteins provide important snapshots in time, longitudinal measurements will further contribute to increasing our molecular understanding of normal physiological development and aging, disease course, and create possibilities for predicting incident disease or mortality. The plasma proteome has been profiled longitudinally in adults, showing high diversity, a strong connection to genetics and relatively high stability[11,12].

[1]Department of Protein Science, KTH Royal Institute of Technology, SciLifeLab, Stockholm, Sweden. [2]Department of Clinical Science and Education, Södersjukhuset, Karolinska Institutet, Stockholm, Sweden. [3]Department of Women's and Children's Health, Karolinska Institutet, Stockholm, Sweden. [4]Sachs' Children and Youth Hospital, Södersjukhuset, Stockholm, Sweden. [5]Institute of Environmental Medicine, Karolinska Institutet, Stockholm, Sweden. [6]Centre for Occupational and Environmental Medicine, Region Stockholm, Stockholm, Sweden. [7]These authors contributed equally: Sofia Bergström, Sophia Björkander. [12]These authors jointly supervised this work: Peter Nilsson, Erik Melén. ✉e-mail: sofia.bjorkander@ki.se

Childhood is a dynamic process, with major developmental changes related to both biological and physiological processes. Some of these processes, for example immune development and puberty, are sex-related or occur during specific time periods[13–15]. Molecular phenotyping using the plasma proteome during healthy childhood and early adulthood may provide important insights into development, including pre- and post-puberty-related alterations. Gaining a fundamental understanding of sex-related differences is a key step in improving the future utilization of proteomic biomarkers in sex-related disease manifestations. Also, characterization of the circulating proteome during healthy childhood could be applied for studying the origin or prediction of diseases throughout the life course[16].

Large-scale studies of the circulating proteome during childhood are few, and the majority are cross-sectional. Two recent studies have highlighted a significant contribution of age, sex, body mass index (BMI), and genetics on the observed proteome in childhood[17,18]. Even fewer studies have longitudinally investigated the circulating proteome during childhood. In a study of 10 healthy children followed from 9 months to 15 years of age, half of the ~2000 measured proteins had age-dependent patterns[19].

In the presented study, we investigated 5416 proteins in plasma obtained longitudinally at four consecutive follow-ups during childhood and early adulthood from 100 randomly selected healthy subjects of a population-based cohort. We aimed to investigate the temporal dynamics of the protein signatures in blood, as well as sex-associated differences, to provide insights into molecular changes during key phases of growth and development.

## Results

We investigated protein profiles in plasma samples from 100 individuals followed longitudinally at four follow-ups during childhood and early adulthood. The levels of 5416 proteins were measured using Olink Explore HT, and the sample set presented here is a subset of the Human Disease Blood Atlas[20].

### Sample demographics and overview

We first compared female and male demographic factors within each follow-up (Table 1). At follow-up 8, female participants had higher median BMI ($p = 0.038$), while the difference in BMI z-scores was not significant ($p = 0.101$). When defining overweight/obesity using sex- and age-specific cut-off values, there was a higher percentage of individuals with overweight/obesity among female participants at follow-up 4 ($p = 0.017$), and a similar pattern was observed for the other follow-ups, although not statistically significant. As expected, the median body fat percentage was higher in female participants at follow-up 24 ($p > 0.001$). The proportions of smokers at follow-ups 16 and 24 were comparable between male participants and female participants, while the percentage of moist snuff users was higher among male participants at the later follow-up ($p = 0.031$). There were, as expected, sex differences in the complete blood cell counts at follow-ups 16 and 24[21].

The age distribution per follow-up is visualized in Fig.1a and shown in Table 1. PCA analysis based on all 3509 proteins that passed our detectability criteria in 97 individuals with complete longitudinal dataset revealed a shift in protein profiles likely driven mainly by age (Fig. 1b), while sex-associated differences were less pronounced, especially before puberty (Fig. 1c).

### Differences in protein levels between consecutive follow-ups

Age-associated univariate differences in the levels of the 3509 proteins were investigated between the three consecutive follow-ups (4 vs 8, 8 vs 16, and 16 vs 24), summarized in volcano plots in Fig. 2a–c. A total of 1879 proteins (54%) were significant, defined as Benjamini–Hochberg adjusted $p$-values below 0.05 together with log2 fold change above 0.5, in at least one comparison between consecutive follow-ups, 179 proteins were significant in all three

comparisons (Fig. 2d). The largest number of proteins that differed between two consecutive follow-ups were observed when comparing follow-up 8 and follow-up 16, with a total of 1416 and 188 proteins with significantly higher and lower levels, respectively, in follow-up 16 (Fig. 2e).

In Fig. 2f, we highlight some of the proteins with the largest differences in either $p$-value or log2 fold change between two consecutive follow-ups and illustrate several different protein level patterns. Complete data on raw and adjusted p-values and log2 fold changes from the univariate comparisons between consecutive follow-ups can be found in Supplementary Data 1 (where a positive fold change indicates higher protein levels at the later follow-up). Proteins that showed a continuous decrease between each consecutive follow-up included VWC2L (von Willebrand factor C domain containing 2 like) which is a developmental protein implicated in neurogenesis, bone differentiation and matrix mineralization, AMBN (ameloblastin) which is involved in mineralization and structural organization of enamel, CD22 (cluster of differentiation 22) which is involved in several aspects of B cell biology and SEZ6 (seizure related 6 homolog) which has been linked to neuronal function[22]. CES2 (carboxylesterase 2), involved in the metabolism of lipids and drugs, decreased from follow-up 4 to follow-up 16, whereas DSPP (dentin sialophosphoprotein) and COL9A1 (collagen type IX alpha 1 chain) which are involved in mineralization of the tooth and a structural component of hyaline cartilage and vitreous of the eye, respectively[22], decreased from follow-up 8 to follow-up 24. We also observed different patterns of proteins with increasing levels from follow-up 4, here exemplified by the two hormones ADM (adrenomedullin) and CCK (cholecystokinin) which are involved in controlling fluid and electrolyte balance and gallbladder contraction and release of pancreatic enzymes in the gut, respectively, and two proteins involved in bone and dentin mineralization and bone turnover: DMP1 (dentin matrix acidic phosphoprotein 1) and CCN5 (cellular communication network factor 5)[22]. Other proteins showed increased levels from follow-up 8 or follow-up 16, for example TRIM25 (tripartite motif containing 25), a protein involved in innate immune defences, MMP3 (matrix metallopeptidase 3) which is involved in the breakdown of extracellular matrix in normal physiological processes as well as in innate immunity, and CELA2A (chymotrypsin like elastase 2 A), a protein implicated in reducing platelet hyperactivation and in regulation of insulin[22]. To replicate the age-associated analyses of Liu et al.[19] and Niu et al.[17], we conducted a longitudinal analysis using a linear mixed-effects model across the four follow-ups. For a total of 350 and 188 overlapping proteins with reported age-related patterns in Liu et al.[19] and Niu et al.[17], respectively, an average of 69% of the age-related associations showed the same direction of association in our dataset. We then performed the age-associated univariate analysis of the levels of the 3509 proteins between the three consecutive follow-ups (4 vs 8, 8 vs 16, and 16 vs 24) separately for female and male participants. This analysis revealed that several proteins had age-related changes between consecutive follow-ups in only one of the sexes. When comparing follow-ups 4 and 8, 234 proteins significantly differed by age only in female participants, and 176 proteins differed by age only in male participants. The corresponding numbers when comparing follow-ups 8 and 16 were 144 only in female participants and 291 only in male participants, and when comparing follow-ups 16 and 24, the corresponding numbers were 668 for female participants and 107 for male participants. Of note, for many of the proteins with significant sex-specific age-associations, there were similar trends for the other sex, without reaching statistical significance (full data displayed in Supplementary Data 2). For visualization, proteins were first filtered on log2 fold change >0.5 and FDR $p$-value < 0.05. We then correlated the log2 fold changes for female participants and male participants for each comparison of consecutive follow-ups (Supplementary Fig. 2).

**Table 1 | Background data on the included 50 female participants and 50 male participants**

| | Follow-up 4 | | | Follow-up 8 | | | Follow-up 16 | | | Follow-up 24 | | |
|---|---|---|---|---|---|---|---|---|---|---|---|---|
| | Female participants | Male participants | p | Female participants | Male participants | p | Female participants | Male participants | p | Female participants | Male participants | p |
| Age in years | 4.3 (4.0–5.1) | 4.3 (4.0–4.6) | 0.542 | 8.4 (7.4–9.3) | 8.3 (7.4–9.4) | 0.185 | 16.8 (16.0–17.9) | 16.7 (15.8–17.9) | 0.367 | 22.6 (21.8–25.0) | 22.5 (21.8–23.7) | 0.572 |
| Weight in kilograms[a] | 18.1 (14.8–26.5) | 17.6 (12.6–23.2) | 0.183 | 30.3 (21.5–44.1) | 27.9 (18.3–43.1) | 0.037 | 60.4 (49.1–85.5) | 68.7 (39.9–93.5) | <0.001 | 63.4 (51.1–108.4) | 75.0 (42.1–109.7) | <0.001 |
| Height in meters[a] | 1.07 (0.98–1.18) | 1.05 (0.95–1.19) | 0.176 | 1.34 (1.20–1.45) | 1.30 (1.17–1.53) | 0.122 | 1.69 (1.56–1.84) | 1.80 (1.55–2.03) | <0.001 | 1.70 (1.58–1.86) | 1.82 (1.61–2.04) | <0.001 |
| BMI[a] | 16.5 (13.9–19.3) | 16.1 (13.6–19.7) | 0.163 | 17.2 (14.1–21.8) | 16.3 (13.5–21.5) | 0.038 | 21.5 (16.5–28.0) | 20.7 (16.6–30.9) | 0.295 | 22.2 (16.6–38.2) | 22.5 (16.3–36.4) | 0.521 |
| BMI z-score[a] | 0.8 (−1.0 to 2.3) | 0.6 (−1.5 to 2.9) | 0.279 | 0.7 (−1.1 to 2.3) | 0.3 (−1.8 to 2.6) | 0.101 | 0.2 (−1.9 to 1.8) | −0.1 (−2.1 to 2.4) | 0.260 | −0.2 (−1.6 to 3.9) | −0.33 (−2.0 to 3.4) | 0.328 |
| BMI categorical (%)[a] Under/normal weight Overweight/obese | 36 (72.0) 14 (28.0) | 45 (91.8) 4 (8.2) | 0.017 | 38 (76.0) 12 (24.0) | 44 (88.0) 6 (12.0) | 0.192 | 39 (78.0) 11 (22.0) | 44 (88.0) 6 (12.0) | 0.287 | 37 (74.0) 13 (26.0) | 39 (78.0) 11 (22.0) | 0.815 |
| Body fat percentage[a] | – | – | | – | – | | – | – | <0.001 | 25.6 (12.8–46.4) | 15.8 (6.3–34.8) | <0.001 |
| Late/post pubertal[a] | – | – | | – | – | | 45 (100.0) | 28 (62.2) | <0.001 | – | – | |
| Cigarette smoker | – | – | | – | – | | 4 (8.3) | 5 (10.0) | 1.000 | 7 (14.0) | 8 (16.0) | 1.000 |
| Moist snuff user | – | – | | – | – | | 1 (2.1) | 2 (4.0) | 1.000 | 4 (8.0) | 13 (26.0) | 0.031 |
| Blood cell counts[a] | | | | | | | | | | | | |
| Basophils ×10⁹/L | – | – | | – | – | | 0.1 (0.1–0.1) | 0.1 (0.1–0.1) | 0.968 | 0.0 (0.0–0.1) | 0.0 (0.0–0.0) | 0.969 |
| Eosinophils ×10⁹/L | – | – | | – | – | | 0.1 (0.1–1.6) | 0.1 (0.1–1.2) | 0.917 | 0.1 (0.0–0.4) | 0.1 (0.0–0.6) | 0.408 |
| Erythrocytes ×10¹²/L | – | – | | – | – | | 4.5 (3.9–5.1) | 5.1 (4.4–6.9) | <0.001 | 4.4 (3.9–5.2) | 5.0 (4.6–7.7) | <0.001 |
| Leukocytes ×10⁹/L | – | – | | – | – | | 6.4 (3.6–10.4) | 5.8 (3.7–10.8) | 0.113 | 6.6 (4.2–14.8) | 5.5 (3.1–11.7) | 0.021 |
| Lymphocytes ×10⁹/L | – | – | | – | – | | 1.9 (1.0–3.4) | 1.9 (1.0–3.8) | 0.467 | 2.0 (1.1–3.8) | 2.0 (1.0–3.2) | 0.112 |
| Monocytes ×10⁹/L | – | – | | – | – | | 0.5 (0.2–0.8) | 0.5 (0.3–1.0) | 0.887 | 0.5 (0.3–0.8) | 0.5 (0.2–0.9) | 0.599 |
| Neutrophils ×10⁹/L | – | – | | – | – | | 3.7 (1.3–6.4) | 3.3 (1.8–6.7) | 0.114 | 3.7 (1.6–11.4) | 2.9 (1.8–9.1) | 0.037 |
| Thrombocytes ×10⁹/L | – | – | | – | – | | 265 (189–345) | 236 (140–430) | 0.028 | 240 (157–352) | 232 (150–385) | 0.349 |
| Hemoglobin (g/L) | – | – | | – | – | | 131 (108–149) | 149 (129–164) | <0.001 | 132 (112–151) | 150 (135–169) | <0.001 |

Continuous values are displayed as median (range), categorical variables are displayed as number (percentage).
Within each follow-up, continuous variables were analysed using the Mann–Whitney U-test (Wilcoxon rank-sum test), categorical variables were analysed using the two-tailed Fisher's exact test.
–: not determined.
[a]One subject lacks weight-, height- and BMI-data from follow-up 4. Two female participants lack body fat percentage from follow-up 24. Five female participants and five male participants lack pubertal status from follow-up 16. One to four subjects lack data on blood cell count at follow-up 16 and/or follow-up 24.

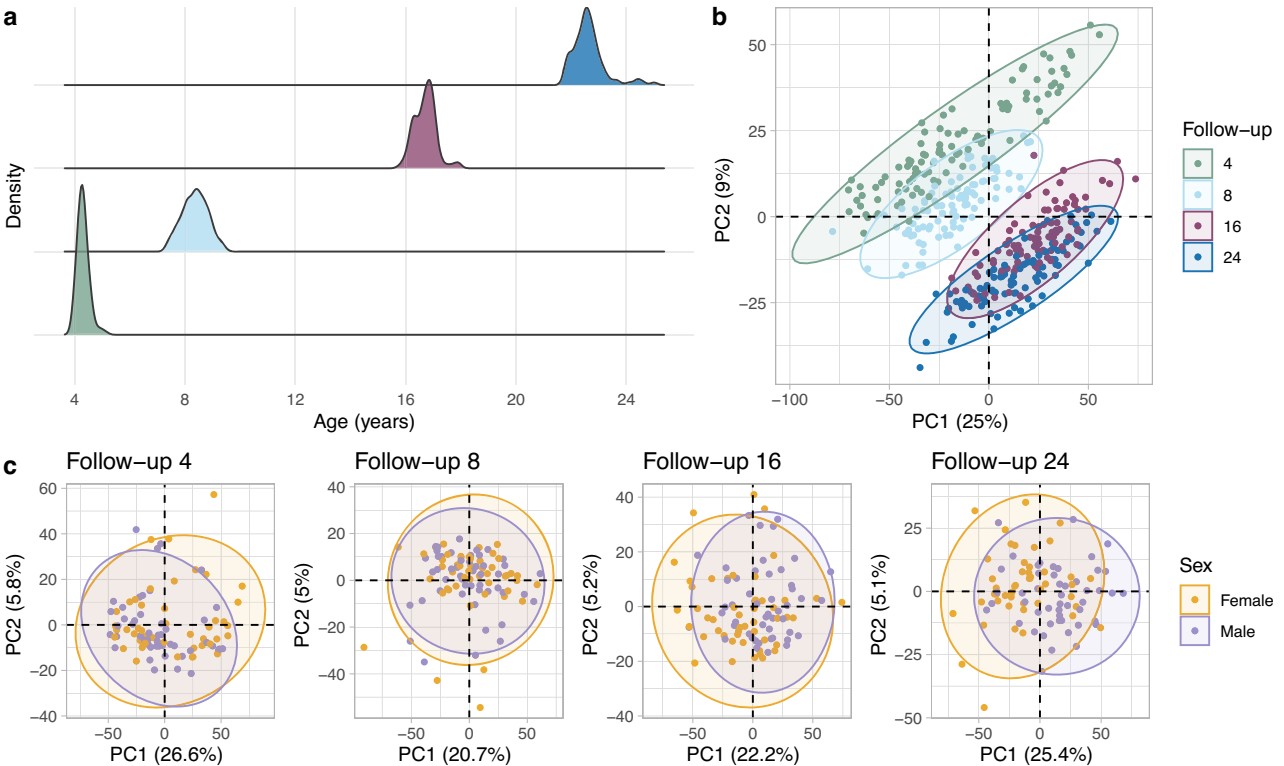

**Fig. 1 | Sample demographics and overview. a** Age distribution per follow-up. **b** PCA based on 3509 proteins, colored by follow-up. **c** PCA based on 3509 proteins per follow-up, colored by sex.

## Trajectory analysis to identify different protein clusters

To obtain groups of proteins with different trajectories, we performed hierarchical clustering on the median longitudinal profiles of the 1879 proteins with at least one significant difference between two consecutive follow-ups. Eight clusters were identified with 68-492 proteins (Fig. 3a), and clusters were further characterized by visualising the number of proteins with significantly higher or lower levels between different follow-ups (Fig. 3b). One representative example protein per cluster was selected to visualise all 97 individuals' trajectories (Fig. 3c). The example was selected among the proteins following the same significance and direction trend as the majority of the proteins in the cluster. Cluster 1 and Cluster 2 were characterized by notable increases from follow-up 8 to 16. Clusters 3 and 8 showed overall decreasing trends from follow-up 4 to 24, while clusters 5 and 7 showed overall increasing trends, with cluster 5 stabilizing between follow-ups 16 and 24 and cluster 7 stabilizing at follow-up 8. Both clusters 4 and 6 displayed decreasing-to-increasing-to-increasing patterns. Data on cluster belonging per protein can be found in Supplementary Data 1 (column M).

Gene Ontology Biological Process (GO BP) enrichment analysis was performed to explore the biological functions associated with the proteins in each cluster. Significant GO terms were identified for all clusters except cluster 5 (Table 2, please see Supplementary Data 3 for a complete list of GO BP terms with raw *p*-values < 0.05 and the proteins related to each term).

Cluster 1 was enriched with GO BP terms related to regulation or modification of protein- or metabolic processes, while cluster 2 was related to intracellular processes and cell cycle. Clusters 3 and 8, with overall decreasing trends, were enriched for terms related to neuron development and neural- and cell projection morphogenesis or regulation of cell secretion, including hormone secretion, respectively, and both were related to cell adhesion. Cluster 7, with overall increasing trends, was related to catabolic and metabolic processes. Finally, the fluctuating clusters 4 and 6 were enriched for terms related

to cellular respiration, immune system, mitochondrial gene expression, chromosome organization, telomere structure, DNA replication, and mRNA processing.

## Sex-associated protein profiles

Next, we investigated sex-associated differences in protein levels within each follow-up. Complete data on raw and adjusted p-values and log2 fold changes can be found in Supplementary Data 4 (where positive fold changes indicate higher protein levels in female participants).

Very few proteins differed significantly between female and male participants at follow-up 4 (AMELY, SPESP1, TXLNGY, XG, and ZP4) and follow-up 8 (AMELY, CRISP2, INSL3, LEP, PUDP, SPESP1, TXLNGY, XG, and ZP4), but we observed more differences at follow-up 16, and the differences were even more prominent at follow-up 24 (Fig. 4a–e). We identified a total of 1111/3509 proteins (32%) with significant differences between male and female participants in at least one follow-up. 183 proteins (5%) were significantly different between sexes at follow-up 16, of which 166 proteins (91%) had higher levels in male participants. At follow-up 24, 1058 proteins (30%) were significantly different between sexes, of which 1030 proteins (97%) had higher levels in male participants. Out of these proteins, 132 (4%) were significantly different between sexes at both follow-ups, 16 and 24 (Fig. 4e), of which 122 proteins (92%) had higher levels in male participants. A PCA analysis based on those 132 proteins at follow-up 24 revealed a clear separation between female and male participants (Fig. 4f). In Fig. 4g, we illustrate some of the proteins that differed by sex. AMELY (amelogenin Y-linked), involved in tooth enamel development and encoded on the Y chromosome[22], was the only protein with a significant difference between sexes exclusively at follow-up 4. Three proteins differed between female and male participants at all four follow-ups: ZP4 (zona pellucida sperm-binding protein 4) was higher in female participants, and SPESP1 (Sperm equatorial segment protein 1) and TXLNGY (taxilin gamma Y-linked) were higher in male participants. ZP4 is involved in

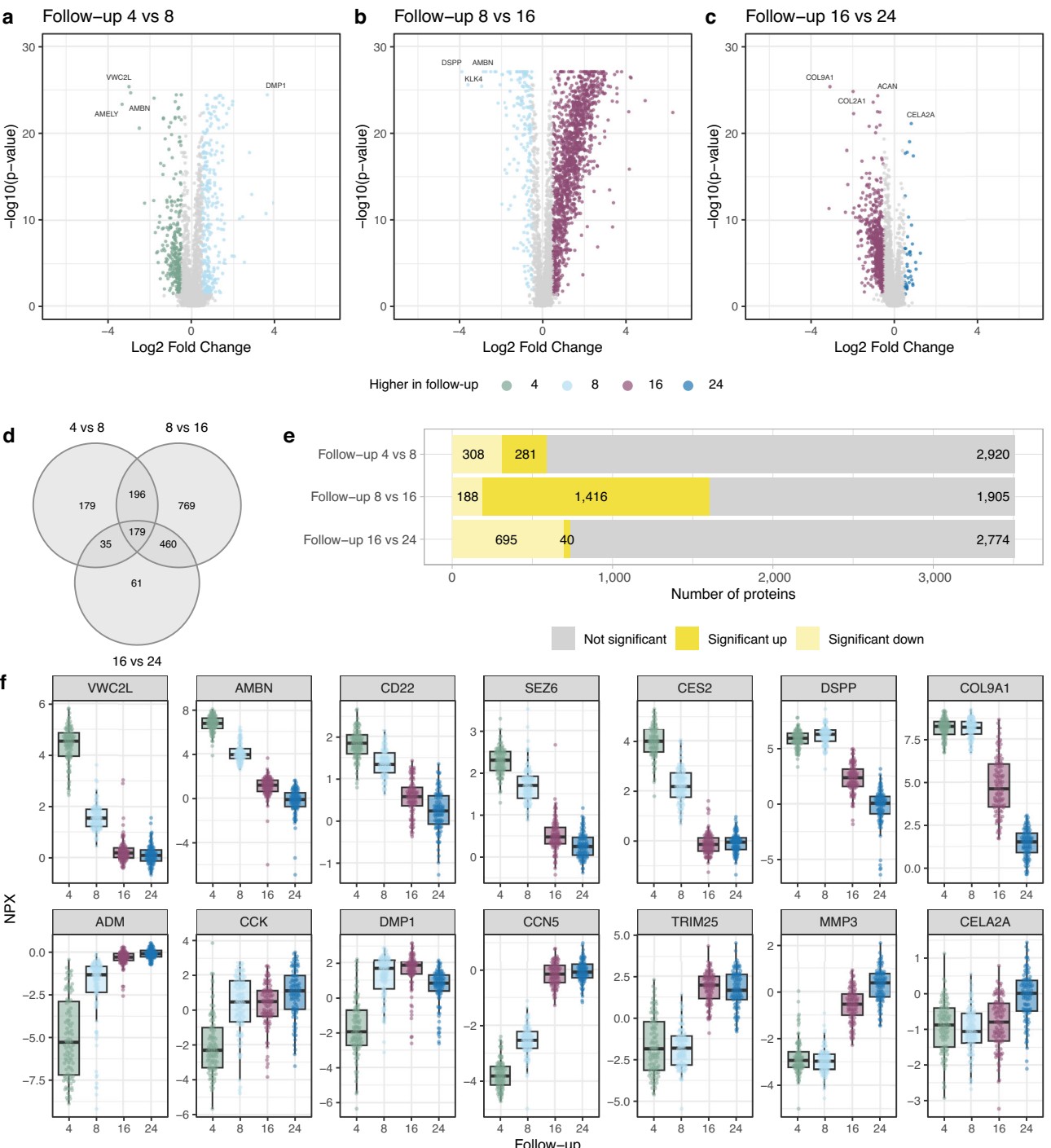

**Fig. 2 | Comparison of protein levels between each follow-up. a–c** Volcano plot summarising age-associated differences between consecutive follow-ups. **d** The overlap of significant proteins between follow-up comparisons. **e** The number of proteins significantly increased or decreased for each follow-up comparison. **f** Examples of fourteen protein profiles (*n* = 97 for each follow-up). For box graphs, the center line denotes the median; the upper and lower bounds of the box represent the 25th and 75th percentiles; the upper and lower whiskers extend to the largest or smallest value no further than 1.5 times the inter quartile range from the box bounds. The two-tailed Wilcoxon signed-rank test corrected for multiple testing using the Benjamini–Hochberg procedure (*p* < 0.05) and a log2 fold change was used to assign statistical significance (**a–c**).

female reproduction, SPESP1 is involved in sperm function, and TXLNGY is suggested to be involved in intracellular vesicle trafficking[22]. XG, encoded on the X-chromosome and a blood group antigen, was higher in female participants at all four follow-ups, but the log2 fold change did not reach the defined cut-off at follow-up 16. As expected, some of the proteins with the strongest association to sex are related to the reproductive system, and the differences manifest primarily at follow-up 16 and/or follow-up 24. The five proteins with lowest *p*-values in both follow-up 16 and 24, all with higher levels in male participants, included EDDM3B (epididymal protein 3B), INSL3 (insulin like 3), SPINT3 (serine peptidase inhibitor, Kunitz type 3), ACRV1 (acrosomal vesicle protein 1) and TEX101 (testis expressed 101), suggested to be involved in testicular function, secreted in the male reproductive system or have possible functions in sperm maturation[22].

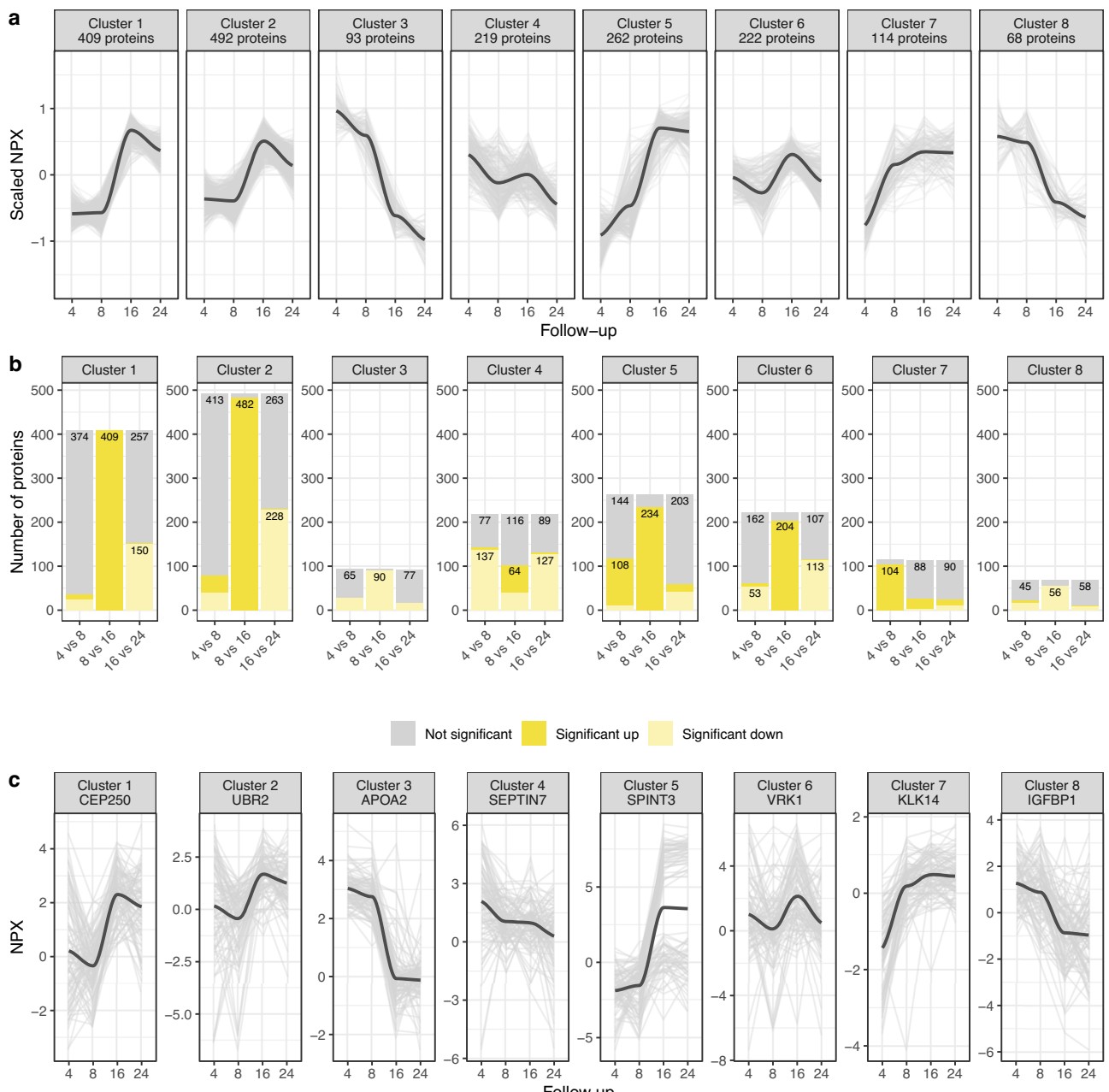

**Fig. 3 | Longitudinal trajectory clustering of protein medians. a** Median long-itudinal protein profiles of scaled NPX values across protein clusters. Each grey line represents the median level of all samples per follow-up per protein. Each black line is a LOESS-smoothed trend line representing the average trajectory across all proteins in one cluster. **b** The number of proteins per cluster with significantly increased or decreased levels, or not significantly different levels, based on the pairwise comparison between consecutive follow-ups. **c** One example protein per cluster. Each black line represents the LOESS-smoothed trend line for the example protein, and the grey lines represent the NPX values for each individual across four follow-ups.

We observed similar patterns for all five proteins, illustrated by INSL3 and SPINT3 in Fig. 4g. In women, we observed higher levels of proteins involved in the female reproductive system, for example PAEP (pro-gestagen associated endometrial protein) and OVGP1 (oviductal gly-coprotein 1), as well as SHBG (sex hormone binding globulin) which regulates the plasma clearance of steroid hormones like testosterone and estradiol[22] at both follow-up 16 and 24. Higher levels of LEP (lep-tin), a crucial regulator of appetite, energy balance, and body weight control that strongly correlates with body fat[22], and GH1 (growth hormone 1), a hormone that stimulates body growth[22], were higher among female participants as compared to male participants at follow-ups 16 and 24, with differences in LEP apparent already at follow-up 8. In relation to growth, males had a delayed drop in the levels of ACAN

(aggrecan), a major component in growth plates[22]. Male and female participants also showed different patterns in the levels of immune-related molecules at follow-ups 16 and 24, here illustrated by the pat-tern recognition receptor SSC4D (scavenger receptor cysteine rich family member with 4 domains/syndecan 4)[22] (Fig. 4g).

Gene Ontology Biological Process (GO BP) enrichment analysis was performed on all proteins with significant differences between sexes at follow-up 16 ($n = 183$) or at follow-up 24 ($n = 1058$) as well as the 132 proteins with significant differences at both follow-ups 16 and 24 (Table 3, please see Supplementary Data 5 for a complete list of GO BP terms with raw $p$-value < 0.05 and the proteins related to each term). The 183 proteins with differences at follow-up 16 were mainly related growth (bone growth and bone development), reproduction

**Table 2 | GO BP enrichment results per age-related cluster**

| Cluster | ID | Description | p-value raw | p-value FDR | Number of proteins |
|---|---|---|---|---|---|
| 1 | GO:0043412 | macromolecule modification | 1.2e-05 | 1.3e-02 | 93 |
| | GO:0036211 | protein modification process | 4.3e-06 | 9.5e-03 | 92 |
| | GO:0051248 | negative regulation of protein metabolic process | 8.1e-05 | 3.9e-02 | 45 |
| | GO:0031400 | negative regulation of protein modification process | 8.6e-05 | 3.9e-02 | 25 |
| | GO:0051246 | regulation of protein metabolic process | 8.9e-05 | 3.9e-02 | 86 |
| 2 | GO:0000082 | G1/S transition of mitotic cell cycle | 1.1e-04 | 3.3e-02 | 17 |
| | GO:0070925 | organelle assembly | 1.2e-04 | 3.3e-02 | 56 |
| | GO:0044772 | mitotic cell cycle phase transition | 1.3e-04 | 3.3e-02 | 27 |
| | GO:0061024 | membrane organization | 1.6e-04 | 3.6e-02 | 55 |
| | GO:2000045 | regulation of G1/S transition of mitotic cell cycle | 1.8e-04 | 3.6e-02 | 15 |
| 3 | GO:0048858 | cell projection morphogenesis | 1.0e-05 | 2.1e-03 | 16 |
| | GO:0048812 | neuron projection morphogenesis | 1.3e-05 | 2.3e-03 | 15 |
| | GO:0007155 | cell adhesion | 1.3e-06 | 7.2e-04 | 30 |
| | GO:0009653 | anatomical structure morphogenesis | 1.5e-05 | 2.4e-03 | 35 |
| | GO:0048699 | generation of neurons | 1.8e-04 | 1.4e-02 | 22 |
| 4 | GO:0045333 | cellular respiration | 1.0e-08 | 2.0e-05 | 14 |
| | GO:0006935 | chemotaxis | 1.1e-04 | 1.7e-02 | 25 |
| | GO:0042330 | taxis | 1.1e-04 | 1.7e-02 | 25 |
| | GO:0061844 | antimicrobial humoral immune response mediated by antimicrobial peptide | 1.2e-04 | 1.7e-02 | 13 |
| | GO:0140053 | mitochondrial gene expression | 1.3e-06 | 6.5e-04 | 8 |
| 6 | GO:0006261 | DNA-templated DNA replication | 1.0e-05 | 6.6e-03 | 8 |
| | GO:0000723 | telomere maintenance | 1.1e-04 | 2.8e-02 | 9 |
| | GO:0032200 | telomere organization | 1.1e-04 | 2.8e-02 | 9 |
| | GO:0006397 | mRNA processing | 1.2e-04 | 2.8e-02 | 17 |
| | GO:0022613 | ribonucleoprotein complex biogenesis | 1.6e-04 | 3.1e-02 | 15 |
| 7 | GO:0170035 | L-amino acid catabolic process | 1.3e-04 | 2.2e-02 | 6 |
| | GO:0044248 | cellular catabolic process | 1.3e-04 | 2.2e-02 | 11 |
| | GO:0019752 | carboxylic acid metabolic process | 1.8e-04 | 2.7e-02 | 17 |
| | GO:1901606 | alpha-amino acid catabolic process | 1.9e-04 | 2.7e-02 | 6 |
| | GO:0043436 | oxoacid metabolic process | 2.3e-04 | 3.0e-02 | 17 |
| 8 | GO:1903532 | positive regulation of secretion by cell | 1.3e-04 | 2.7e-02 | 9 |
| | GO:0098609 | cell-cell adhesion | 1.5e-04 | 2.7e-02 | 15 |
| | GO:0046887 | positive regulation of hormone secretion | 1.6e-04 | 2.7e-02 | 6 |
| | GO:0007155 | cell adhesion | 1.8e-05 | 1.2e-02 | 22 |
| | GO:0051047 | positive regulation of secretion | 1.9e-04 | 2.7e-02 | 9 |

Five significant GO BP terms are shown per cluster. For the over-representation analysis, a one-sided version of Fisher's exact test, corrected for multiple testing using the Benjamini–Hochberg procedure ($p < 0.05$), was used to assign statistical significance.

(sperm-egg recognition, binding of sperm to zona pellucida) and hormone regulation (steroid hormone secretion, response to estradiol, endocrine hormone section) while the 132 proteins with significantly different levels at both follow-ups 16 and 24 were mainly related to reproductive processes, including negative regulation of reproductive stress, as well as response to different stimuli (light, activity, UV and ethanol). In contrast, the 1058 proteins with differences at follow-up 24 were most prominently related to translation, autophagy, catabolic processes, and intracellular transport.

## Sex-related differences in proteins after adjustment for covariates

Next, linear regression models were used to investigate sex differences after adjustment for covariates. The first model, including BMI z-scores, smoking, and moist snuff usage, was used to investigate proteins with observed sex-associated differences at follow-up 16, follow-up 24, or at both follow-ups 16 and 24. Out of the 132 proteins with sex differences at both follow-ups 16 and 24, all proteins remained significant at follow-up 16, and 130 proteins remained significant at follow-up 24 after adjustment for these factors. Further, all 183 proteins except two with sex differences at follow-up 16 remained significant after adjustment, and 1023 out of 1058 proteins with sex differences at follow-up 24 remained significant. LEP, PROK1 (prokineticin 1) and SHBG were the only proteins with significant association to BMI z-scores in the first model at both follow-ups 16 and 24, and the correlations between the levels of these proteins and BMI z-scores are visualised in Fig. 5a. All three proteins were however still found to be associated with sex after adjusting for BMI z-scores, moist snuff usage and smoking. In summary, adjustment based on BMI z-scores, smoking, or moist snuff use attenuated only a few of the observed differences between sexes. When exchanging BMI z-scores to body fat percentage (only available at follow-up 24) in the second model, 938 proteins remained significant, and 22 proteins were found to be significantly associated with body fat percentage (Supplementary Data 6). Correlations between eight proteins, selected based on lowest p-values, and body fat percentage are visualised in Fig. 5b. Out of the 22

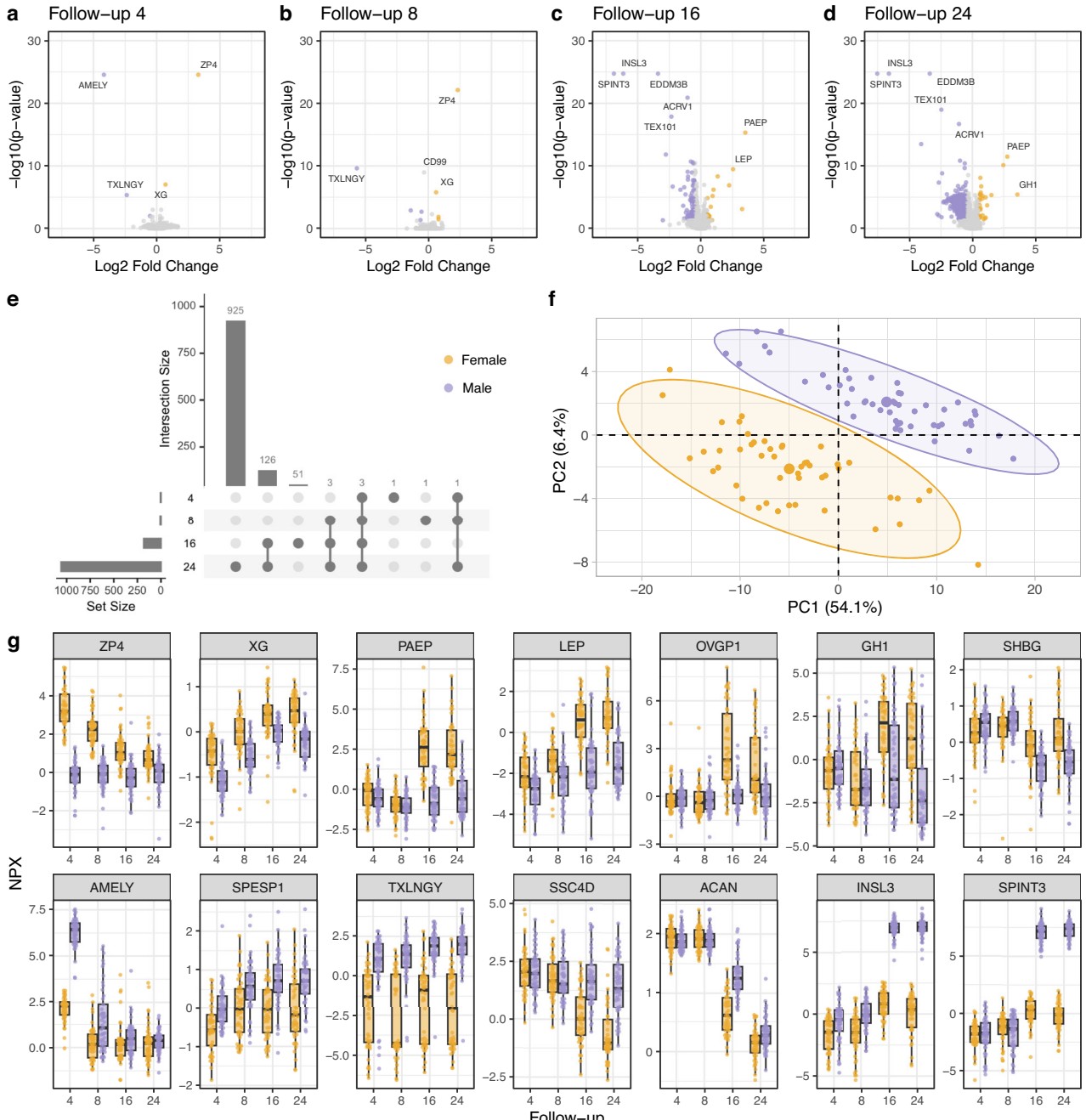

**Fig. 4 | Sex-associated protein profiles. a–d** Volcano plots summarising the sex-associated differences per follow-up. **e** Upset plot visualizing the number of significant proteins per follow-up and combinations of follow-ups. The horizontal bars represent the number of significant proteins per follow-up, and the vertical bars represent the size of each intersection. **f** PCA plot for the subjects in follow-up 24 based on 132 proteins with a significant difference in both follow-up 16 and 24. **g** Examples of fourteen protein profiles (*n* = 48 for female participants (yellow) and *n* = 49 for male participants (purple) at each follow-up). For box graphs, the center line denotes the median; the upper and lower bounds of the box represent the 25th and 75th percentiles; the upper and lower whiskers extend to the largest or smallest value no further than 1.5 times the inter quartile range from the box bounds. The two-tailed Wilcoxon rank sum test corrected for multiple testing using the Benjamini–Hochberg procedure (*p* < 0.05) and a log2 fold change was used to assign statistical significance (**a**–**d**).

proteins found to be significantly associated with body fat percentage, 14 proteins remained significantly associated with sex after adjustments (Supplementary Data 6).

Furthermore, we had the unique opportunity to adjust for erythrocyte and leukocyte counts given the sex-related differences in red and white blood cells (Table 1). At follow-up 16, 147 out of the 183 proteins (80%) remained significantly different between sexes, while at follow-up 24, a larger proportion of the protein differences were attenuated, with 640 out of the 1058 proteins (60%) still having a

significant difference between female and male participants after adjustment. Out of the 132 proteins with a significant difference between sexes at both follow-ups 16 and 24, 107 and 105 proteins remained significant (~80%) at follow-ups 16 and follow-up 24, respectively, after adjusting for blood cell counts.

## Discussion
This population-based study aimed to investigate the temporal dynamics of the blood proteome from childhood into early adulthood

**Table 3 | GO BP enrichment results for proteins that differed by sex at follow-up 16 or at follow-up 24, or at both follow-ups 16 and 24**

| Protein set | ID | Description | *p*-value raw | *p*-value FDR | Number of proteins |
|---|---|---|---|---|---|
| Significant proteins at follow-up 16 (*n* = 183) | GO:0035036 | Sperm-egg recognition | 4.6e-05 | 6.2e-02 | 6 |
| | GO:0014823 | Response to activity | 7.2e-05 | 6.2e-02 | 8 |
| | GO:0007339 | Binding of sperm to zona pellucida | 8.5e-05 | 6.2e-02 | 5 |
| | GO:0032355 | Response to estradiol | 1.0e-04 | 6.2e-02 | 9 |
| | GO:0044060 | Regulation of endocrine process | 1.6e-04 | 8.2e-02 | 6 |
| | GO:0035929 | Steroid hormone secretion | 2.4e-04 | 8.6e-02 | 5 |
| | GO:2000831 | Regulation of steroid hormone secretion | 2.4e-04 | 8.6e-02 | 5 |
| | GO:0098868 | Bone growth | 3.8e-04 | 1.2e-01 | 5 |
| | GO:0060986 | Endocrine hormone secretion | 4.5e-04 | 1.2e-01 | 6 |
| | GO:0060348 | Bone development | 4.7e-04 | 1.2e-01 | 11 |
| Significant proteins at follow-ups 16 and 24 (*n* = 132) | GO:0035036 | Sperm-egg recognition | 6.3e-06 | 1.4e-02 | 6 |
| | GO:0007339 | Binding of sperm to zona pellucida | 1.6e-05 | 1.8e-02 | 5 |
| | GO:0032355 | Response to estradiol | 5.3e-05 | 4.0e-02 | 8 |
| | GO:0009988 | Cell-cell recognition | 1.6e-04 | 8.9e-02 | 6 |
| | GO:0009416 | Response to light stimulus | 3.6e-04 | 1.6e-01 | 9 |
| | GO:0014823 | Response to activity | 5.1e-04 | 1.8e-01 | 6 |
| | GO:0006749 | Glutathione metabolic process | 5.6e-04 | 1.8e-01 | 5 |
| | GO:0009411 | Response to UV | 6.4e-04 | 1.8e-01 | 7 |
| | GO:2000242 | Negative regulation of reproductive process | 1.4e-03 | 3.4e-01 | 5 |
| | GO:0045471 | Response to ethanol | 1.5e-03 | 3.4e-01 | 6 |
| Significant proteins at follow-up 24 (*n* = 1058) | GO:0006412 | Translation | 9.8e-12 | 3.2e-08 | 71 |
| | GO:0006413 | Translational initiation | 1.0e-09 | 1.6e-06 | 30 |
| | GO:0006914 | Autophagy | 1.2e-08 | 7.0e-06 | 68 |
| | GO:0061919 | Process utilizing autophagic mechanism | 1.2e-08 | 7.0e-06 | 68 |
| | GO:0016236 | Macroautophagy | 1.4e-08 | 7.0e-06 | 48 |
| | GO:0009894 | Regulation of catabolic process | 1.5e-08 | 7.0e-06 | 113 |
| | GO:0002181 | Cytoplasmic translation | 1.8e-08 | 7.0e-06 | 19 |
| | GO:0046907 | Intracellular transport | 2.1e-08 | 7.0e-06 | 131 |
| | GO:0019941 | Modification-dependent protein catabolic process | 2.2e-08 | 7.0e-06 | 62 |
| | GO:0043632 | Modification-dependent macromolecule catabolic process | 2.2e-08 | 7.0e-06 | 62 |

The ten GO BP terms with lowest raw *p*-values are shown per protein set. For the over-representation analysis, a one-sided version of Fisher's exact test, corrected for multiple testing using the Benjamini-Hochberg procedure (*p* < 0.05), was used to assign statistical significance.

to provide insights into molecular changes during growth and development. Overall, we found that age had a significant impact on the measured proteins and that sex-related effects became prominent during adolescence and early adulthood.

We observed significant differences between two consecutive follow-ups (4 vs 8, 8 vs 16, or 16 vs 24) for 1879 out of the 3509 proteins that passed our detectability criteria (54%), with sets of proteins that seem to be affected at different age transitions. 179 proteins were significantly different in all three comparisons. Further, we observed both increasing, decreasing, and fluctuating patterns, reflecting the many dynamic physiological changes that occur during the first two decades of human development. We observed most changes between follow-ups 8 and 16, when the majority of changing proteins were up-regulated (1416/1604, 88%). 584 of these 1416 proteins were then down-regulated between follow-ups 16 and 24, indicating that these fluctuating age-related patterns are likely linked to major changes related to growth and puberty. Age is a strong determinant of circulating protein profiles, as was recently shown by Niu et al. in a large cross-sectional study including two independent cohorts of children and adolescents, with 40% of the >1000 measured proteins found to be associated with age[17]. Longitudinal protein profiling data during childhood are, however, limited. Liu et al. observed that more than 50% of 1747 proteins measured with mass spectrometry had age dependent

profiles, based on 10 individuals followed at nine timepoints from >1 to 15 years of age[19]. Stockfelt et al. measured 230 proteins in plasma from children followed longitudinally from birth to 8 years of age with the aim to predict allergy development. Using PCA, they identified two clusters, where plasma samples from cord blood and one month of age clustered together and separated from the later time points[23]. Mikus et al. analysed 97 proteins in plasma longitudinally from 6 months to 5 years of age and found that most proteins were significantly altered by age[24].

From available studies, the proportion of age-related proteins appears to be relatively high in children, likely even more prominent in studies with a longitudinal design. This proportion appears to vary more for adult cohorts, which is likely linked to the varied age span of included subjects, methods used, and p-value thresholds chosen[4,5,25].

To get an overview of the biology of age-related proteins, we clustered the 1879 identified age-associated proteins and observed eight clusters with varying patterns, of which seven clusters were significantly enriched for specific GO-terms. Clusters 3 and 8 with decreasing protein age profiles, were found to be enriched for proteins connected to cell adhesion, and cluster 7, with an increasing age profile was enriched for proteins related to catabolic processes with similar observations in the study by Liu et al.[19]. Further, we found the sharply

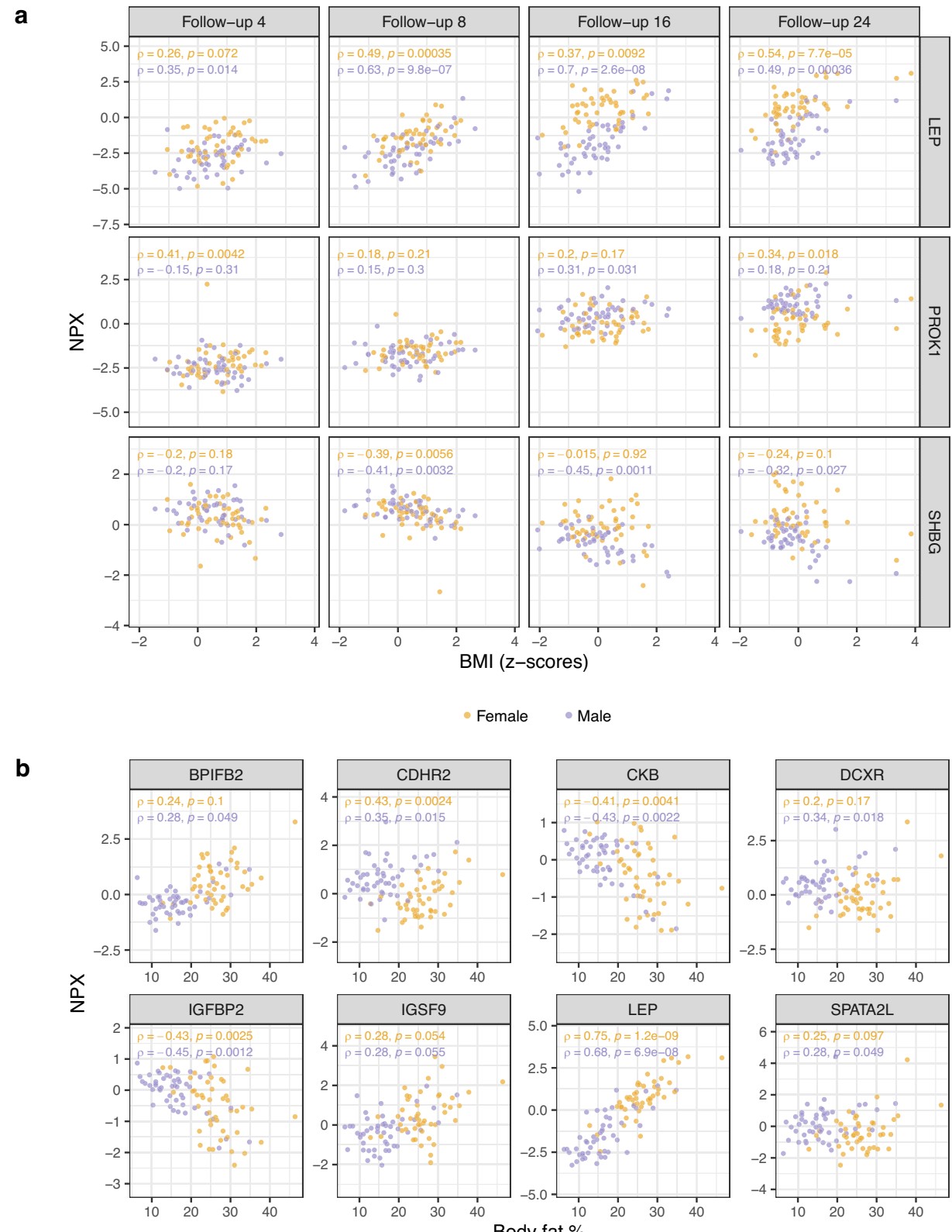

**Fig. 5 | Spearman correlation between proteins and BMI z-scores or body fat percentage. a** Correlations between LEP, PROK1, or SHBG and BMI z-scores per follow-up. **b** Correlations between BPIFB2, CDHR2, CKB, DCXR, IGFBP2, IGSF9, LEP or SPATA2L and body fat percentage at follow-up 24. The two-tailed Spearman correlation test was used to assign statistical significance.

decreasing cluster 3 to be enriched for proteins related to neural development, generation and morphogenesis, which is in line with the rapid brain development that occur during the first 5 years of life[26]. Among the proteins showing the strongest difference between follow-ups, we found proteins related to the formation of bone, cartilage, and teeth (for example, VWC2L, AMBN, and COL9A1). Furthermore, we observed differences in several immune-related proteins, for example, decreasing levels of soluble CD22 (SIGLEC-2), normally expressed on the surface of mature B cells, where it regulates activation and function and is a promising target in B cell malignancies[27,28]. The different branches of the immune system dynamically change and develop during childhood[29], and we believe that our study will contribute to increased understanding of this intricate and finely tuned process. Interestingly, we found notable age-related decreases in the proteins SEZ6 (seizure related 6 homolog) and SEZ6L (seizure related 6 homolog like), two proteins implicated in development and function of the nervous system. These proteins have been associated with neurological and psychiatric disorders like Alzheimer's disease and autism spectrum disorders, as well as febrile seizures, a condition that affects children[30–37]. The highest levels of CES2 and CES3 (carboxylesterase 2 and 3) were observed at follow-up 4, followed by a gradual decrease. Considering that these proteins are involved in the breakdown of toxins and drugs, our results might contribute to the understanding of the elimination of such compounds during childhood.

Sex is a strong driver of circulating protein levels, and gaining a fundamental understanding of sex-related differences is one key to improving future utilization of proteomic biomarkers. However, studies describing sex differences in longitudinal protein profiling during childhood remain limited. We identified 1111/3509 proteins (32%) with different levels in female participants compared to male participants in at least one follow-up. There were only a few proteins that differed between sexes at follow-up 4 and follow-up 8, whereas 5% and 30% differed between sexes at follow-up 16 and 24, respectively. The total number of sex-related proteins observed by us is in line with data presented by Niu et al., where cross-sectional data on >1000 proteins were obtained from >3000 children and adolescents aged 5–20 years showed that 32% of proteins were related to sex after adjustment for age[17]. Koprulu et al. recently analysed protein levels in plasma from two large-scale population-based cohorts of adults, the Fenland study and UK Biobank, using the SOMAscan assay (v4) (4775 proteins) and Olink Explore 3072 (2923 proteins) respectively, and found most targets (69%) to have significant sex differences, where 768 (42%) overlapped with concordant results in both assays[38]. We speculate that the lower percentage observed in our study is linked to our young study population. The circulating proteome associates with factors enriched in adult populations that will likely show deviations depending on sex, for example, high BMI and body fat percentage, and comorbidities like cardiometabolic and autoimmune diseases, conditions that are not as frequently manifested in children and young adults. We also included fewer subjects, likely masking some sex-related differences. We observed that >90% of the proteins with different levels between sexes at follow-ups 16 and/or 24 had higher levels in male participants compared to female participants. Koprulu et al. also identified more proteins to be higher in male participants (62–64%)[38]. Our results further demonstrate age-related alterations in biological processes related to proteins that differed by sex. GO BP terms related to reproduction were evident when investigating proteins related to sex at either follow-up 16 or follow-ups 16 and 24. Processes related to bone growth were also evident for proteins with sex differences at follow-up 16. In opposite, for proteins that differed by sex at follow-up 24, the most prominent terms were related to translation, autophagy, catabolic processes, and intracellular transport. Of note, at follow-up 24, proteins related to reproduction and hormones were also evident (Supplementary Data 5).

Not surprisingly, the most prominently different proteins between sexes are involved in different aspects of reproduction. We also observed higher levels of AMELY, encoded on the Y chromosome, in male participants compared to female participants at follow-up 4. The unexpectedly high levels in female participants at follow-up 4 are likely caused by cross-reactivity with another member of the amelogenin family, namely AMELX (not included in this analysis). It is well known that there are sex-related differences in the immune system[13,14], and we identified several immune-related proteins that differed with sex. For example, the levels of the scavenger receptor SSC4D, involved in pattern recognition and opsonization[39], showed a clear age-related drop in female participants, not evident in male participants.

Next, we explored how available potential confounding factors contributed to the observed sex-related differences. The circulating proteome has previously been connected to BMI and body composition[40–43], and even certain proteins in the cord blood proteome have been found to associate with birth weight and early-life growth variables[44]. We did not observe a strong impact of BMI z-scores on our results. Further, adjusting for body fat percentage attenuated 12% of the observed sex-related differences at follow-up 24. Female participants had higher levels of the appetite- and energy balance-regulating hormone LEP, evident at all follow-ups but only significant at follow-ups 16 and 24. We also observed a strong correlation between BMI z-scores or body fat percentage and LEP in both sexes. In adults, LEP shows a strong correlation to female sex, which is associated with puberty and with the higher percentage of body fat among female participants[45–48]. Leptin levels in childhood have also been shown to predict adiposity gain in children without obesity[49], highlighting that this is an important factor to study during childhood. Furthermore, we had the unique opportunity to investigate how adjustment for blood leukocyte and erythrocyte counts affected sex-associated proteins at follow-ups 16 and 24. Adjustment for these factors attenuated a notable proportion (approximately 40%) of the observed sex-related differences at follow-up 24, but not at follow-up 16. This might indicate that proteins associated with sex only at follow-up 24 are related to processes that are more closely linked to immune- and haematological processes. These associations have been described in adults previously[50,51], but require more investigation, particularly in pediatric studies. Further, if blood cell counts are also associated with age-related differences in protein levels throughout the life course remains to be determined.

Characterization of the circulating proteome during healthy childhood could provide a powerful tool for detection and prediction of diseases throughout the life course[16]. The main aim of this study was not to identify proteins associated with complex diseases, but to describe how age and sex influence part of the circulating proteome during healthy childhood. The informative and predictive value of cross-sectional and longitudinal protein profiling has been demonstrated as a valuable tool to increase our understanding of disease courses and mechanisms[52–54]. In children, the longitudinal serum proteome profile was found to predict the onset of type-1 diabetes in children with susceptebility[55] and protein signatures were recently shown to be linked to cardiometabolic conditions like obesity, insulin resistance, and hypertension, and could predict steatotic liver disease[18]. Proteomic aging and proteomic aging clocks are an established representative of health in adults and[3,5,25]. Lehallier et al. observed distinct waves of proteomic changes throughout adult life that were linked[2]. Future work will tell if diseases across the life-course can be predicted already during childhood using protein-based biomarkers in combination with lifestyle and clinical data.

Our study highlights the importance of taking sex into consideration also when investigating protein biomarkers in children, previously highlighted for adults in, for example, cardiovascular disease and depression disorders[10,56–58]. The same applies to BMI and

body fat percentage, where our results align with earlier reports identifying it as an important factor to consider in proteomic studies[43], especially for conditions and comorbidities linked to overweight and obesity.

This study has some limitations that should be considered. Olink Explore HT is highly multiplex and enables the analysis of over 5000 proteins, however, does not cover the whole proteome, and complementing the data with other methods would give an even broader protein coverage. Due to the explorative nature of the Olink Explore HT panel[20], not all 5416 proteins were detected in a sufficient number of samples, and we took an inclusive approach and filtered the dataset to remove proteins with less than 10 samples (10%) above LOD at all four follow-ups. The longitudinal design of the cohort enables the analysis of proteins in the same individuals over time, however, it unavoidably leads to different storage times for samples from different follow-ups, and we cannot rule out that the levels of some proteins are affected by this factor. Further, sample handling and lab equipment may have changed throughout the study. We also acknowledge that this cohort is relatively small, with mainly white participants.

In conclusion, our population-based longitudinal study demonstrates dynamic age- and sex-related changes in protein levels during the first two decades of life. The observed age-dependent profiles for a majority of the measured proteins highlight the importance of careful consideration of age in pediatric studies, and our study further elucidates at what age-span some of the sex differences observed in adults start to occur. The current study deepens the understanding of the proteomic perspective of human development from childhood to adulthood, and the potential and limitations of protein profiling in disease etiology and for future protein biomarker research.

## Methods

### The human disease blood atlas

The data presented here is a subset of the data generated within the Human Disease Blood Atlas project as a part of the Human Protein Atlas (www.proteinatlas.org). The Human Disease Blood Atlas aims to create a comprehensive map of protein levels in human blood across major diseases, and to create an open-access resource that contributes with valuable insights to the global proteomics community. The first phase of the Human Disease Blood Atlas, including measurements of up to 5416 proteins in health and across 59 diseases, has been previously published[20,59].

### Ethics approval and consent to participate

This study was conducted in accordance with the Declaration of Helsinki and was approved by the Swedish Ethical Review Authority and the Regional Ethics Review Board in Stockholm (Approval 2016/1380-31/2). All participants and/or their legal guardian(s) gave written informed consent to participate in the study.

### The study population

The study subjects included here are part of the Swedish population-based cohort BAMSE (Barn[Child], Allergy, Milieu, Stockholm, Epidemiology)[60]. Briefly, all children who were born in four pre-defined municipalities in Stockholm between 1994 and 1996 were invited to participate ($n = 7221$), out of which 5488 children were eligible (exclusion: planned move, insufficient knowledge in the Swedish language, serious illness of the child or siblings already included). Out of all eligible subjects, 4089 agreed to participate (>95% of white descent[61]), constituting the original cohort. The study subjects have been followed with repeated questionnaires (1-, 2-, 4-, 8-, 12-, 16-, 24-year follow-ups and COVID-19 follow-up phases 1-4) and extensive clinical examinations (4-, 8-, 16-, 24-year follow-ups and COVID-19 follow-up phase 2)[62,63].

### Subject selection

For this project, potential participants from the original BAMSE cohort were identified based on the availability of longitudinal plasma samples at follow-ups 4, 8, 16, and 24, as well as clinical data ($n = 583$). Exclusion criteria were pregnancy or missing data on smoking, moist snuff usage, or BMI at follow-up 24. Out of the 583 eligible participants, 100 subjects (men $n = 50$, women $n = 50$) were randomly selected. Venous blood was collected in EDTA tubes at the clinical examinations, and plasma was obtained by centrifugation, aliquoted, and stored at −80° Celsius.

### Definition of descriptive and clinical variables

During the clinical examinations at follow-ups 4, 8, and 16, weight was measured by regular weight scales, while weight and body fat percentage were measured using a Tanita MC 780 body composition monitor during follow-up 24. Height was measured by a wall-mounted stadiometer at all follow-ups. BMI was calculated as weight in kilograms divided by the square of height in meters ($kg/m^2$). BMI was standardized into z-scores using the World Health Organization child growth standards for ages 0–5 years[64] and growth reference data for those aged 5–19 years[65]. At follow-up 24, BMI was transformed into sex-specific BMI z-scores based on the observed values within the full BAMSE cohort[66]. At follow-ups 4, 8, and 16, overweight/obesity was defined using sex- and age-specific cut-off values for BMI[67] and at follow-up 24 as BMI ≥ 25.0. Pubertal status was defined as previously described[63]. Data on current (daily or occasional) smoking and moist snuff use were taken from self-reported questionnaire data at follow-ups 16 and 24. Complete blood cell counts were measured in whole blood acquired from venipuncture by routine flow cytometry at the Karolinska University Laboratory in Stockholm[21]. The descriptive and clinical variables are summarized by sex in Table 1.

### Measurements of protein levels

5416 proteins were measured in plasma samples using Olink Explore HT, which combines the Proximity Extension Assay technology with antibody-binding capabilities coupled with next-generation sequencing (NGS) readout. Initially, the samples (40 µl) were distributed in 96-well plates in a constrained randomized fashion together with other samples from the Human Disease Blood Atlas based on cohort, diagnosis, age, and sex. Ten wells per plate were dedicated to Olink controls, including two negative controls (buffer), five plate controls (pool of plasma originating from healthy blood donors), and three sample controls (pool of plasma used to assess inter and intra-precision). All samples from the BAMSE cohort were analysed in the same batch, and all samples from the same subjects were randomized in the same plate. The assays were divided into eight different dilution blocks, with between 68 and 1314 protein assays per block. The data obtained was reported as relative protein quantification using the Normalized Protein eXpression (NPX) unit. The NPX values were obtained by adjusting the counts from the sequencing data based on the extension control per sample and protein assay, followed by log2-transformation. The NPX values were subsequently normalized using intensity normalization, which normalizes the NPX values based on the median levels of each protein across all samples per plate.

### Data analysis

Demographic variables were analysed within each follow-up with STATA SE (version 16). Continuous variables were analysed using the Wilcoxon rank-sum test, and categorical variables were analysed using Fisher's exact- or chi²-test.

The open-source software R (version 4.5.1)[68] was used for data processing, main statistical analysis, and visualizations. Limit of detection (LOD) per protein assay and plate was calculated using the

negative Olink controls (buffer run as a normal sample) using the OlinkAnalyze package (*version 4.3.1*)[69] with the LOD method set to "NCLOD". Proteins with at least 10% of the samples above LOD in at least one follow-up were kept in the dataset. Two proteins were included in three different dilution blocks, and the measurements from the block with the highest number of samples above LOD were kept in the dataset. Following the recommendation from the provider, all proteins from dilution block 8, corresponding to the most abundant proteins (*n* = 68, dilution 1:100,000, <1.3% of the total library content), were excluded from the analyses due to technical issues where a subset of bases may be miscalled when utilizing the NovaSeq X with the 10B flow cell, which can sporadically lead to increased coefficients of variation across assays within block 8. Three individuals had sample quality control warnings for 270-1204 proteins each, and those individuals were excluded from the dataset to get complete longitudinal data, no values were imputed. Next, we assessed if there were any samples deviating from the rest of the cohort based on dimensionality reduction, but no samples were determined as outliers. After the curation, the data consisted of measurements of 3509 proteins in 97 individuals across all four follow-ups (388 samples). A total of *n* = 1194 proteins (22%) were detected above LOD in all 388 samples, and each individual had, on average *n* = 1993 proteins (37%) above LOD in all four follow-ups. Since Olink Explore is a discovery/exploratory platform with a predefined set of proteins of which a large proportion is intracellular and not expected to be in the circulation, we do not expect full detectability in all types of biological sample sources, including plasma. The detectability in the here presented study is in line with previously published plasma studies using Olink Explore panels[20,70].

Principal Component Analysis (PCA) was performed on scaled and centered data (*factoextra::fviz_pca_ind, version 1.0.7*)[71] both based on all 388 samples and per follow-up. The potential univariate differences of protein levels between different sample groups were evaluated using the Wilcoxon rank-sum test (*stats::wilcox.test*) for unpaired comparisons or the Wilcoxon signed-rank test for paired comparisons, and corrected for multiple testing using the Benjamini-Hochberg procedure (*stats::p.adjust(method = "BH")*). The results were summarized together with the difference in median protein levels per group in volcano plots. Adjusted p-values below 0.05 together with a log2 fold change of 0.5 were considered significant. Upset plots were used to visualize intersections between different sets of proteins (*UpSetR:: upset, version 1.4.0*)[72].

For clustering of the longitudinal protein trajectories, the data were scaled and centered before the median was calculated per protein and follow-up. Proteins with at least one significant difference between two consecutive follow-ups were included (*n* = 1879), and hierarchical clustering was performed based on Euclidean distance using Ward's clustering (*stats::hclust(method = "ward.D2")*). The optimal number of clusters was selected using gap statistics, the silhouette method (*factoextra::fviz_nbclust(method = "gap_stat" or method = "silhouette"), version 1.0.7*)[71] and c-index (*NbClust:: NbClust(index = "cindex", version 3.0.1*)[73], together with bootstrap-based cluster stability analysis (*fpc::hclustCBI and fpc::clusterboot, version 2.2-13*)[74]. An eight cluster solution provided the best tradeoff between cluster stability and biological interpretability (Supplementary Fig. 1), with average Jaccard stability index of cluster 1:0.47, cluster 2: 0.59, cluster 3: 0.85, cluster 4: 0.70, cluster 5: 0.50, cluster 6: 0.64, cluster 7: 0.53, cluster 8: 0.65.

Gene ontology Biological Process (GO BP) enrichment analysis (*clusterProfiler::enrichGO, version 4.16.0*)[75] was performed on the proteins in each cluster defined from the 1879 proteins with observed age-associated differences. The 1879 proteins were used as background, and annotations were sourced from the org.Hs.eg.db database (*version 3.21.0*)[76]. GO BP enrichment analysis was also performed on identified sex-associated proteins, with all 3509 proteins as background. GO BP

terms with Benjamini-Hochberg adjusted p-values below 0.05 were considered significant. To assess sex-associated differences in protein levels while accounting for potential confounding factors, linear regression analysis (*stats::lm*) was performed. The first model included BMI z-scores, moist snuff usage, and smoking as covariates (follow-up 16 or follow-up 24), while BMI z-scores were exchanged for body fat percentages in the second model (only follow-up 24). The third model included erythrocyte and leukocyte counts (follow-up 16 or follow-up 24). Furthermore, we assessed if proteins with observed sex-related differences were significantly associated with BMI and or body fat percentage based on the linear regression models and visualised significant proteins with scatterplots and Spearman correlations. Benjamini–Hochberg adjusted *p*-values below 0.05 were considered significant.

To replicate the age-associated analyses of Liu et al.[19] and Niu et al.[17], we conducted a longitudinal analysis using a linear mixed-effects model across the four follow-ups (*lme4:: lme, version 1.1.37*). To obtain *p*-values for the fixed effects, we used (*lmerTest, version 3.1.3*). Protein expression data were normalized using rank-based inverse normal transformation, and sex was included as a covariate. Protein-level results from the supplementary materials of the original studies were used for replication. For Liu et al.[19], the supplementary table reports the results from two mixed-effects models: one including a linear age term and another including both linear and quadratic age terms. As our analysis was based on a linear mixed-effects model, we used the linear age coefficients reported by Liu et al.[19]. Proteins with a nominal *p* value < 0.05 were considered significant, yielding 735 proteins, of which 350 overlapped with the proteins available in our dataset.

### Reporting summary

Further information on research design is available in the Nature Portfolio Reporting Summary linked to this article.

## Data availability

All data associated with this study are available within this paper, its supplementary information, or in the Source data, or may be obtained upon request from the corresponding author. Due to consent and legal constraints, the individual-level data cannot be deposited openly. Access to individual-level data can be granted for both non-commercial research and validation purposes, and upon request to the corresponding author. A request should contain the following: the name of PI and host organization, contact details, the scientific purpose of the data access request, the commitment to inform when the data has been used in a publication, the commitment not to host or share the data outside the requesting organization, and a statement of non-commercial use of data. Data access requests will be responded to within four weeks unless ethical constraints require new ethical permits, and the data will be available for a mutually decided timeframe. Source data are provided with this paper Source data are provided with this paper.

## Code availability

All code necessary for the data analyses and visualization is available at https://github.com/sofiabergstrom/childhood-protein-profiling[77].

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

## Acknowledgements

First, we would like to thank all the BAMSE participants and BAMSE staff for contributing to the study. In addition, we would like to thank the entire staff of the Human Protein Atlas for their efforts, and especially the Human Disease Blood Atlas team. SciLifeLab Affinity Proteomics Infrastructure Unit Stockholm, supported with sample handling, preparations, and randomisations, and SciLifeLab Affinity Proteomics Infrastructure Unit Uppsala and the National Genomic Infrastructure Uppsala (NGI), generated the Olink data and provided assistance in protein analyses. The Human Protein Atlas with the Human Disease Blood Atlas was supported by the WCPR grant for Knut and Alice Wallenberg Foundation (KAW 2022.0318) to MU, and the SciLifeLab & Wallenberg Data Driven Life Science Program (KAW 2020.0239) to FE. The BAMSE cohort study was supported by the Swedish Research Council (2019-01060; 2020-02170; 2024-03164) to EM, The Swedish Research Council for Health, Working Life and Welfare (2017-00526) to AB, Formas (2016-01646) to AB, The Swedish Heart-Lung Foundation to EM, the European Union (European Research Council, TRIBAL No 757919) to EM, Region Stockholm (ALF project) to EM, and the Asthma and Allergy Research Foundation to IK. H.D. was supported by LillaBarnetsFond, Spädbarnsfonden (Swedish infant death foundation), and Region Stockholm.

## Author contributions

S.Be., S.Bj., S.K., M.U., P.N., and E.M. conceptualized this study. S.Bj., A.B., I.K., A.S.M., and E.M. contributed to BAMSE study design and sample collection. S.Be., F.E., M.U., and P.N. organised the data generation. SBe. and SBj. performed the data analysis and generated figures, tables, and visualizations with support from M.B.A., S.K.M., H.D., and S.K. P.N. and E.M. supervised the project. S.Be. and S.Bj. wrote the manuscript. All authors revised, read, and approved the final manuscript.

## Funding

## Competing interests

The authors declare no competing interests.
