## [Transparent Peer Review file · Nature Communications]

Longitudinal protein profiling of blood during childhood into early adulthood

Corresponding Author: Dr Sophia Björkander

Version 0:

Reviewer comments:

Reviewer #1

(Remarks to the Author)

The paper by Bergström et al. aims to characterize age-related changes and sex differences in plasma protein levels of 100 participants at four timepoints during childhood and early adulthood. The authors observe strong and widespread age-related changes at all timepoints and sex differences after puberty that are partly explained by differences in blood cell proportions. The statistical methodology applied in this work is simple, but sound. This study provides a useful resource of proteins that change their levels throughout childhood, although the consequences of these differences for diseases and other phenotypes are not investigated.

Major comments:

- 1) It would be interesting to know if any of the observed age differences are sex-specific - that is if there are proteins that change with age at the later follow-ups only in one of the sexes.
- 2) The fact that adjusting for cell type percentages removes much of the sex differences in protein levels raises the concern that part of the observed age differences in protein levels may also be due to age-related differences in cell counts.
- 3) Do you have an explanation for why more proteins are up-regulated than down-regulated from follow-up 8 to 16 and more - down-regulated than up-regulated from 16 to 24 years?

Minor comments:

- 1) There is only one p-value column per comparison in the supplementary table, and as far as I understand this is the BH-corrected p-value. Please also include the raw uncorrected p-values.
- 2) If I understood the text correctly, clustering was done on median values per protein per follow-up. Then it is a bit misleading to call this clustering on the "longitudinal protein profiles" in the main text, because it does not take into account individual longitudinal trajectories. Please make it more clear in the Results section.
- 3) What was the rationale to choose BH-corrected $p < 0.001$ as the significance cut-off?
- 4) How many negative controls were available for LOD calculation? Olink recommends using fixed LOD values for datasets with fewer than 10 negative controls.

Reviewer #2

(Remarks to the Author)

Bergstorm et al., performed plasma proteome profiling using the Olink HT Explore panel in 100 individuals sampled longitudinally at four time points (ages 4, 8, 16, and 24 years) from a population-based Swedish cohort. The authors assessed the effects of age and sex on the circulating proteome, identifying 2,700 age-associated proteins out of 3,509 passing QC, which could be grouped into five distinct trajectory clusters. They further reported 908 sex-associated proteins, most of which emerged at ages 16 and 24—after the onset of puberty—with higher levels predominantly observed in males.

Some of these sex-related differences were partially explained by secondary traits such as BMI, body fat percentage, and blood cell counts.

Overall, the manuscript is well written and presents an important longitudinal analysis that could significantly advance our understanding of dynamic proteomic changes from childhood through early adulthood. However, the study would benefit from additional analyses and clarifications to enhance its interpretability and impact.

Major:

1. The authors highlight representative proteins within each age-dependent trajectory cluster. It would strengthen the study to comment on how well the dataset captures well-characterized proteins with known developmental patterns (e.g., growth factors, hormonal proteins). How do these trajectories compare to reported trajectories in the previous studies?
2. Given that approximately one-quarter of proteins differ between males and females at ages 16 and 24, sex-stratified longitudinal trajectory clustering would likely increase the resolution and interpretability of the patterns. Have the authors considered this? Furthermore, please clarify whether sex was included as a covariate when identifying age-dependent proteins.
3. The study would benefit from an integrative framework to jointly model the effects of age, sex, BMI (or body fat percentage), and their interactions. Such modeling could uncover more complex interaction effects that may be masked in the current univariate analyses.
4. The manuscript states that “all proteins from dilution block 8, corresponding to the most abundant proteins, were excluded due to technical issues.” For interested readers, please elaborate on the nature of these technical issues.
5. The statement “As expected, not all 5416 proteins were detected in a sufficient number of samples...” may not be intuitive to all readers. Please clarify why this is expected, e.g., due to assay sensitivity limits or biological absence in plasma? If possible, please reference supporting data from published studies.

Minor:

1. Please specify centrifugation conditions during plasma processing (centrifugal force, duration, and temperature).
2. Please include the number of QC'ed proteins in the abstract for clarity.
3. The Benjamini–Hochberg adjusted p-value threshold of 0.001 is quite stringent. How many significant associations would be observed under more conventional thresholds (e.g., 0.01 or 0.05)?
4. In Table 2, consider listing the proteins assigned to each ontology term to maximize the interpretive value of this dataset.
5. The phrase “males had a slower drop in the levels of ACAN...” is ambiguous and should be clarified. I suggest replacing “slower” with “delayed” or “occurring at a later age” unless the authors specifically meant the rate of decline in protein levels per unit of time.
6. A typo in Line 435 “processed”
7. Line 463, the comment “however, age stratified analysis was not performed” was not entirely correct. The authors did adjust for age when testing for sex effects. The point about stage-specific differences is valid, but the sentence should be corrected to reflect the actual analysis.

Reviewer #3

(Remarks to the Author)

This manuscript presents a longitudinal proteomic study of 100 individuals across four developmental stages (ages 4, 8, 16, and 24), profiling over 5,000 plasma proteins using the Olink Explore HT platform. It offers a valuable data resource on age- and sex-related dynamics in the circulating proteome during childhood and early adulthood—a relatively underexplored period. While the dataset is rich, several analytical and interpretative aspects require further clarification and methodological rigor before the manuscript is suitable for publication. In particular, the distinction between biological effects (age, sex) and technical confounders, as well as the rationale for certain statistical analyses and interpretations, should be strengthened. Specific concerns are outlined below:

1. Age vs. batch effects in PCA

In Figure 1B, the clear separation of samples by follow-up time is interpreted as a “dramatic age-associated shift in protein profiles.” However, given the longitudinal design, variation in sample collection years, and the use of multiplex assays, age is likely highly confounded with technical factors such as plate/batch and storage time. I recommend that the authors (i) provide a more detailed description of how samples from different follow-ups were randomized and distributed across plates/batches, and (ii) explicitly assess and correct for potential batch effects (e.g., by including batch as a covariate in linear models or applying a dedicated batch-correction algorithm). Quantitative evidence demonstrating that the PCA separation is not primarily driven by batch effects would strongly support the conclusion that these shifts reflect true age-related proteomic changes.

2. Justification for the definition of age-differential proteins

The authors define age-differential proteins using a Benjamini-Hochberg adjusted p-value < 0.001 for inter-group comparisons, without specifying whether a fold change threshold was applied. The volcano plots in Figures 2A–C also suggest that no such threshold was used, which appears inconsistent with standard stringent criteria for defining differential expression. I recommend adopting more robust filtering criteria, such as incorporating a specific fold change threshold (e.g., $|FC| > 1.5$ or 2) alongside the adjusted p-value, to improve the reliability of the identified age-associated proteins.

3. Justification and robustness of the 5 trajectory clusters

The authors clustered 2,700 age-associated proteins into five longitudinal trajectory clusters using hierarchical clustering with Ward's method. Although gap statistics and the silhouette method are mentioned, no actual evidence supporting the

choice of five clusters is provided. To improve transparency, I suggest that the authors (i) include gap statistic and/or average silhouette plots for different cluster numbers (e.g., in the Supplementary Material), and (ii) briefly comment on within-cluster homogeneity or cluster stability. This would help readers assess the robustness and biological relevance of the identified trajectory clusters.

4. Interpretation of the sharp increase in sex-associated proteins at 16/24 years

The authors report almost no sex-associated proteins at ages 4 and 8, followed by 104 and 857 at ages 16 and 24, respectively. However, Figure 1C suggests some sex-related separation across all follow-ups. It would be helpful to clarify whether the near-absence of sex-associated proteins at younger ages reflects a true biological pattern or limited statistical power and/or the stringent multiple testing threshold (adj. $p < 0.001$). For instance, do canonical X/Y-linked proteins show consistent effect sizes at earlier ages but fail to meet the significance threshold? A brief sensitivity analysis (e.g., using a slightly relaxed FDR threshold or reporting effect-size distributions) could help reconcile the PCA patterns with the univariate results.

5. Lack of hormone-related GO enrichment at 24 years

At age 24, when subjects are fully sexually mature, one might expect sex-associated proteins to be enriched in pathways related to sex steroid hormones or reproductive signaling. However, the GO enrichment results mainly highlight translation, proteasomal catabolism, and protein folding, with no obvious hormone-related terms. I suggest that the authors (i) comment on the coverage of hormone-related proteins on the Olink Explore HT panel and (ii) discuss why classical hormone-related pathways are not prominently enriched. Clarifying whether this reflects a biological pattern, limitations in GO annotation, or the choice of background set and significance thresholds would be valuable.

6. Interpretation of covariate-adjusted models as “factors contributing to sex differences”

In the section titled “Factors contributing to the observed differences between females and males,” the authors fit linear models adjusting for BMI z-scores, body fat percentage, smoking, snuff use, and blood cell counts, then report how many proteins remain sex-associated. Conceptually, these models assess the robustness of sex differences to covariate adjustment but do not formally quantify the contribution or mediation of these factors. I recommend (i) rephrasing this section to emphasize covariate adjustment and robustness (e.g., “Sex differences after adjustment for BMI and body composition”) rather than “factors contributing,” and (ii) avoiding causal language unless formal mediation or variance decomposition analyses are performed. If the goal is to evaluate mediation explicitly, additional analyses are needed.

7. Comparison with existing pediatric/adolescent proteomics datasets to assess generalizability

The study focuses on a single birth cohort (BAMSE) and a specific platform (Olink Explore HT). While many reported patterns—such as the high proportion of age-associated proteins and the sharp rise in sex-differential proteins during adolescence—are biologically plausible, it is unclear whether they are cohort- or platform-specific or general features of the developing plasma proteome. I strongly encourage the authors to contextualize their findings more explicitly with existing pediatric/adolescent proteomics datasets (e.g., Liu et al. 2017, Mikus et al. 2021, Niu et al. 2025) and, if feasible, perform at least a partial cross-cohort comparison. For example, they could:– Examine whether key age- and sex-associated proteins show consistent direction and effect sizes in published datasets;– Comment on whether the main trajectory patterns (e.g., increasing, decreasing, pubertal peaks) are conserved across cohorts or specific to BAMSE/Olink. Such analyses would help clarify the robustness and generalizability of the reported signatures.

8. Experimental validation of key findings

Although the Olink dataset provides valuable descriptive insights, the current conclusions rely entirely on statistical comparisons. For biologically important findings—such as representative age-related trajectories or pronounced sex-associated differences—experimental validation is essential. I recommend that the authors select several key proteins (e.g., those most strongly associated with age or sex) and verify their relative abundance using an orthogonal method, such as ELISA or Western blot, in a subset of samples. This would strengthen confidence in the quantitative accuracy of the Olink results and help ensure that the observed differences are not platform- or batch-specific artifacts.

Version 1:

Reviewer comments:

Reviewer #1

(Remarks to the Author)

I thank the authors for their clarifications and text updates. I do not have any other comments.

Reviewer #2

(Remarks to the Author)

The authors have addressed my previous concerns.

The authors stated "Access to codes can be obtained upon request to the corresponding author."

For transparent and reproducible research, I strongly recommend the authors deposit the code in a public repository such as Zenodo, or else state the reasons why code cannot be shared.

Reviewer #3

(Remarks to the Author)

The authors have provided thorough and convincing responses to my prior comments. This study constitutes a tour de force in the field and deserves prompt publication.

(Remarks to the Author)

The authors have provided thorough and convincing responses to my prior comments. This study constitutes a tour de force in the field and deserves prompt publication.

Response to Reviewers for the manuscript:

"Longitudinal protein profiling of blood during childhood into early adulthood" by Bergström S/Björkander S et al.

Reviewer 1

The paper by Bergström et al. aims to characterize age-related changes and sex differences in plasma protein levels of 100 participants at four timepoints during childhood and early adulthood. The authors observe strong and widespread age-related changes at all timepoints and sex differences after puberty that are partly explained by differences in blood cell proportions. The statistical methodology applied in this work is simple, but sound. This study provides a useful resource of proteins that change their levels throughout childhood, although the consequences of these differences for diseases and other phenotypes are not investigated.

Reply: *We thank the reviewer for the helpful comments and have revised the manuscript accordingly. We believe that these results have future implications for increased understanding of diseases and other phenotypes, and such studies are ongoing but were not within the scope of this study.*

Major comments:

1) It would be interesting to know if any of the observed age differences are sex-specific - that is if there are proteins that change with age at the later follow-ups only in one of the sexes.

Reply: *We agree with the reviewer that this type of analysis would give added value to the study. We have therefore performed the age-associated analysis by comparing the protein levels between two consecutive follow-ups separately in females and males. We now display these data in new Supplementary Table 2, where we show the raw p-values, FDR p-values and log₂ fold change, and in new Supplementary Figure 2 where we show the correlations of log₂ fold changes for males and females for each comparison (follow-ups 4 vs 8, 8 vs 16 and 16 vs 24). This analysis revealed that several proteins significantly changed with age only in one of the sexes. Of note, there are no proteins that show an opposite age-pattern in the different sexes, for example increased levels in females and decreased levels in males when comparing the same two follow-ups. These results are now described in the Results (pages 13-14, lines 275-288).*

2) The fact that adjusting for cell type percentages removes much of the sex differences in protein levels raises the concern that part of the observed age differences in protein levels may also be due to age-related differences in cell counts.

Reply: *We agree with the reviewer that differences in blood cell counts with age may partly relate to some of the observed age-related differences. However, we consider it being a reflection of the still unexplored biological relevance of blood cell counts in proteomic research. Due to the lack of comparative published studies, we have not elaborated on this data extensively in the manuscript but believe that it is relevant and transparent to display the data. In this cohort, we only have blood cell count data available at follow-ups 16 and 24. **Table R1** below displays a comparison between blood cell counts at these two follow-ups. For females, we found differences in eosinophil, erythrocyte and thrombocyte counts, and for males there were only differences in eosinophil and erythrocytes counts. Blood cell counts vary during childhood, for example, it is known that platelet count generally drops during childhood. Since we do not have access to the data from all four follow-ups, we cannot*

address this matter even though we agree with the reviewer that it would be interesting. We have added a sentence to the Discussion pointing out the need to address this matter in future studies (page 24-25, lines 556-558).

Table R1. Comparison of blood cell counts between follow-up 16 and follow-up 24, separately for females and males.

Variable	P-value females (Wilcoxon matched-pairs signed rank test)	P-value males (Wilcoxon matched-pairs signed rank test)
Basophils x10 ⁹ /L	ND	ND
Eosinophils x10 ⁹ /L	0.0017	0.0210
Erythrocytes x10 ¹² /L	0.0096	0.0916
Leukocytes x10 ⁹ /L	0.4180	0.9858
Lymphocytes x10 ⁹ /L	0.1074	0.6379
Monocytes x10 ⁹ /L	0.3547	0.8056
Neutrophils x10 ⁹ /L	0.6951	0.8766
Thrombocytes x10 ⁹ /L	0.0009	0.2353
Hemoglobin (g/L)	0.8599	0.2578

ND=could not be determined due to differences in reported values below the limit of detection in the two different follow-ups.

3) Do you have an explanation for why more proteins are up-regulated than down-regulated from follow-up 8 to 16 and more - down-regulated than up-regulated from 16 to 24 years?

Reply: This is an interesting observation. We speculate that this is mainly related to the profound changes related to both puberty and growth, that take place between ages 8 and 16, and that some of these proteins then down-regulate after growth and puberty are completed (by age 24). Out of the 1416 proteins that have higher levels at follow-up 16 compared to follow-up 8, 584 proteins were then down-regulated at follow-up 24 out of a total of 695 down-regulated proteins, showing that a majority of proteins that were down-regulated between follow-ups 16 and 24, had been upregulated between follow-up 8 and 16, supporting their relation to growth and puberty. We have now highlighted this in the Discussion on page 20, lines 439-442.

Minor comments:

1) There is only one p-value column per comparison in the supplementary table, and as far as I understand this is the BH-corrected p-value. Please also include the raw uncorrected p-values.

Reply: We have added columns with raw p-values in all tables.

2) If I understood the text correctly, clustering was done on median values per protein per follow-up. Then it is a bit misleading to call this clustering on the "longitudinal protein profiles" in the main text, because it does not take into account individual longitudinal trajectories. Please make it more clear in the Results section.

Reply: Thank you for highlighting this. We have now added the word "median" at page 14, line 292 and to the Figure 3 legend to show that clustering was performed on median values.

3) What was the rationale to choose BH-corrected $p < 0.001$ as the significance cut-off?

Reply: In line with comments from the other reviewers, we have altered the cut-off to classify statistical significance. The new cut-off is BH-corrected $p < 0.05$ AND a log2 fold change of

>0.5. This has somewhat altered the results throughout the manuscript; however, it has not affected the main overall results or the example proteins included in Figures 2 and 4 with the exceptions of CD99 that has been exchanged for SPESP1 in Figure 4g and slight alterations in proteins displayed in Figure 5.

4) How many negative controls were available for LOD calculation? Olink recommends using fixed LOD values for datasets with fewer than 10 negative controls.

Reply: *LODs were based on 56 negative controls (2 per plate for a total of 28 plates), and therefore, fixed LODs are not used.*

Reviewer 2

Bergstorm et al., performed plasma proteome profiling using the Olink HT Explore panel in 100 individuals sampled longitudinally at four time points (ages 4, 8, 16, and 24 years) from a population-based Swedish cohort. The authors assessed the effects of age and sex on the circulating proteome, identifying 2,700 age-associated proteins out of 3,509 passing QC, which could be grouped into five distinct trajectory clusters. They further reported 908 sex-associated proteins, most of which emerged at ages 16 and 24—after the onset of puberty—with higher levels predominantly observed in males. Some of these sex-related differences were partially explained by secondary traits such as BMI, body fat percentage, and blood cell counts.

Overall, the manuscript is well written and presents an important longitudinal analysis that could significantly advance our understanding of dynamic proteomic changes from childhood through early adulthood. However, the study would benefit from additional analyses and clarifications to enhance its interpretability and impact.

Reply: *We thank the reviewer for the helpful comments and have revised the manuscript accordingly.*

Major:

1. The authors highlight representative proteins within each age-dependent trajectory cluster. It would strengthen the study to comment on how well the dataset captures well-characterized proteins with known developmental patterns (e.g., growth factors, hormonal proteins). How do these trajectories compare to reported trajectories in the previous studies?

Reply: *Thank you for the relevant comment. By using the Protein Class annotation in the Human Protein Atlas, we assessed the coverages of growth factors and hormones in the Olink Explore HT platform. We found that for 149 annotated growth factors, 99 were included in the Explore platform, of which 79 proteins were included based on our LOD cut-off. For hormones, we found 99 annotated proteins of which 47 were included in the platform and 45 included based on our LOD cut-off (**Figure R1**). For age- and sex-related plots of these proteins, please see **Figure R2** for hormones and **Figure R3** for growth factors. Further, in Supplementary Table 1, there is a column assigning each protein to a cluster, based on the age-associated analysis.*

*There is, to our knowledge, a lack of defined values for different pediatric age groups and a limited number of corresponding and comparable studies. We used two existing datasets offering opportunities for comparisons: Liu CW et al J Proteomics 2017 (DOI: 10.1016/j.jprot.2016.11.016) and Niu L et al Nat Genet 2025 (DOI: 10.1038/s41588-025-02089-2). For the 79 growth factors and 45 hormones (some of which are overlapping), a total of 31 proteins were reported in one or both of these studies. The overall comparability between our study and these two studies is high with very few opposite results (**Table R2** below), showing that our dataset capture both growth factors and hormones, of which many where age- and or sex-related. Of note, the studies by Liu et al. (2017) and Niu et al. (2025) are both different from our study: In Liu et al. (2017), they included only 10 children followed longitudinally between 9 months and 15 years of age, and in the cross-sectional study by Niu et al. (2025) they included 3000 children and young adults ages 5-20 years. Hence, platform/method, analysis and power differ significantly between these studies. We have not elaborated further on growth factors and hormones in the manuscript due to the very limited number of comparable results.*

Growth factors (n = 149)

Hormones (n = 99)

Figure R1. Growth factors (n=149) and Hormones (n=99) were annotated based on the Protein Class function in the Human Protein Atlas. Circles show the proportions of non-included proteins (grey) and included proteins split as Detected (turquoise green) and Not detected (moss green) based on our defined cut offs. Growth factors: 53% Detected, 13% Not detected, 34% not included. Hormones: 45% Detected, 2% Not detected, 53% not included.

Figure R2. 45 detected hormones were annotated based on the Protein Class function in the Human Protein Atlas that are covered in the Olink HT panel. Green=males, purple=females

Figure R3. 79 detected growth factors were annotated based on the Protein Class function in the Human Protein Atlas that are covered in the Olink HT panel. Green=males, purple=females

Table R2. Age- and sex-related patterns of hormones and growth factors also measured in either Niu et al. (2025) and/or Liu et al. (2017).

Protein	Age-pattern in our study	Sex-pattern in our study	Niu et al. (2025)	Liu et al. (2017)
BGLAP	Increase 4-8, decrease 8-24	Higher in males 16	ND	Increase-then-decrease
BMP10	Decrease 4-16	-	ND	Flat
CLEC11A	Decrease 8-24	-	ND	Flat
CSF1	-	-	ND	Flat
EFEMP1	Increase 4-16	-	Increase age	Increase
FGFBP2	Decrease 8-24	Higher in males 16	Decrease age, higher in males	Increase-then-decrease
GDF2	Decrease 4-16	-	ND	Flat
GMFG	Increase 4-16	Higher in males 24	ND	Increase-then-decrease
IGFBP1	Decrease 8-16	-	Decrease age, higher in males	Decrease
IGFBP2	Decrease 8-16	Higher in males 24	Decrease age, higher in males	Curved decrease
IGFBP3	-	-	Increase age, higher in females	Increase

IGFBP4	Increase 4-16	-	Increase age, higher in females	Curved increase
IGFBP6	Increase 4-16	-	Increase age, higher in females	Decrease-then-increase
IGFBP7	-	-	Higher in males	Flat
INHBC	-	-	Increase age, higher in females	Decrease-then-increase
LEFTY2	Increase 4-8	-	No association	ND
LTBP2			ND	Flat
MANF	Increase 8-16	Higher in males 24	ND	Increase-then-decrease
MSTN	Increase 4-16	Higher in males 16 & 24	ND	Increase
OGN	-	-	Decrease with age, higher in females	Decrease-then-increase
PDGFA	Increase 8-16, decrease 16-24	Higher in males 24	ND	Decrease
PDGFB	Increase 8-16, decrease 16-24	Higher in males 24	ND	Flat
PDGFD	Decrease 4-8, increase 8-16, decrease 16-24	-	ND	Decrease
STC2	Increase 4-16	-	ND	Curved increase
TFF1	-	-	ND	Decrease
THBS4	Decrease 8-16	Higher in males 16	Decrease with age, higher in males	Decrease-then-increase
TIMP1	-	-	Increase with age	Decrease
TYMP	Increase 8-16	Higher in males 24	ND	Decrease
VEGFC	Decrease 16-24	Higher in males 24	ND	Missing pattern
VGF	Decrease 8-16	-	ND	Decrease
TG	Increase 8-16	Higher in males 16	ND	Missing pattern

2. Given that approximately one-quarter of proteins differ between males and females at ages 16 and 24, sex-stratified longitudinal trajectory clustering would likely increase the resolution and interpretability of the patterns. Have the authors considered this? Furthermore, please clarify whether sex was included as a covariate when identifying age-dependent proteins.

Reply: We agree with the reviewer that this type of analysis would give added value to the study. We have therefore performed the age-associated analysis by comparing the protein levels between two consecutive follow-ups separately in females and males. We now display these data in new Supplementary Table 2, where we show the raw *p*-values, FDR *p*-values and log₂ fold change, and in new Supplementary Figure 2 where we show the correlations of log₂ fold changes for males and females for each comparison (follow-ups 4 vs 8, 8 vs 16 and 16 vs 24). This analysis revealed that several proteins significantly changed with age only in one of the sexes. Of note, there are no proteins that show an opposite age-pattern in the different sexes, for example with increased levels in females and decreased in males when comparing the same two follow-ups. These results are now described in the Results (pages 13-14, lines 275-288). Differences in protein levels between two consecutive follow-ups were evaluated using the Wilcoxon signed-rank test, hence sex was not adjusted for in the primary age-related analysis.

3. The study would benefit from an integrative framework to jointly model the effects of age, sex, BMI (or body fat percentage), and their interactions. Such modeling could uncover more complex interaction effects that may be masked in the current univariate analyses.

Reply: For this paper, we have chosen to perform pair-wise comparisons between consecutive follow-ups. Even though we agree with the reviewer that joint modelling can

reveal more complex interactions, for this paper we have decided to keep the current line of analysis for several reasons:

1. Previous analyses as presented by *Álvez et al, Science 2025*, attempted interaction analyses with age, sex, BMI on protein profiles but the results did not give meaningful insights about underlying biology.
2. We believe that our current line of analysis is easier to interpret for the general reader and is more suitable for the present paper.
3. The other reviewers have suggested sex-stratified analysis in relation to age; hence this is the approach we have now chosen to incorporate in the manuscript.

4. The manuscript states that “all proteins from dilution block 8, corresponding to the most abundant proteins, were excluded due to technical issues.” For interested readers, please elaborate on the nature of these technical issues.

Reply: We have elaborated more on the exclusion of proteins in block 8 in *Methods* (page 8, lines 151-156).

5. The statement “As expected, not all 5416 proteins were detected in a sufficient number of samples...” may not be intuitive to all readers. Please clarify why this is expected, e.g., due to assay sensitivity limits or biological absence in plasma? If possible, please reference supporting data from published studies.

Reply: Thank you for highlighting this. As indicated by the reviewer, not all proteins are present at all times in plasma in concentrations high enough to be detected by the assay. Since Olink Explore HT is a discovery/exploratory platform with a predefined set of proteins of which a large proportion is intracellular and not expected to be in the circulation, we do not expect full detectability in all types of biological sample sources, including plasma. Hence, it is likely that biological absence in plasma is the main cause of lack of detection. As shown in *Álvez et al Science 2025*, in the longitudinal Wellness study cohort of healthy adults, on average, each individual had $n=2014/5416$ proteins (37%) consistently detected above LOD in all visits. In BAMSE, each individual had on average $n=1993$ proteins (37%) above LOD at all follow-ups, and $n=1194$ proteins (22%) were above LOD in all 388 samples included in the analysis. In the current study, we included all proteins for which at least 10% of the samples had levels above LOD in at least one of the four follow-ups, resulting in 3509 proteins. In the UK-Biobank, plasma from almost 50,000 subjects has been profiled by the Olink HT Explore 3072, measuring $n=2941$ proteins (*Sun BB et al Nature 2023 DOI: 10.1038/s41586-023-06592-6*). This study does not state how many proteins were detected in 100 percent of the samples but described that $n=787/2941$ proteins (27%) had 30% or more of the samples below LOD, confirming that there is a proportion of proteins that is not detected in plasma using the Olink Explore assays.

In the manuscript, we have tried to further describe the assay detectability. Firstly, in the limitations section (page 26, line 585), the phrasing “as expected” has been changed to “Due to the explorative nature of the Olink Explore HT panel (*Alvez et al Science 2025*)”. In the *Methods* (page 8, lines 162-168), we now state: “A total of $n=1194$ proteins (22%) were detected above LOD in all 388 samples, and each individual had on average $n=1993$ proteins (37%) above LOD in all four follow-ups. Since Olink Explore are discovery/exploratory platforms with a predefined set of proteins of which a large proportion is intracellular and not expected to be in the circulation, we do not expect full detectability in all types of biological sample sources, including plasma. The detectability in the here presented study is in line with previously published plasma studies using Olink Explore panels (*Alvez et al Science 2025, Sun et al Nature 2023*).”

Minor:

1. Please specify centrifugation conditions during plasma processing (centrifugal force, duration, and temperature).

Reply: *For all follow-ups, EDTA-tubes were centrifuged at 2000*g for 10 minutes at room temperature before plasma was retrieved by pipetting and stored at -80 degrees Celcius. We have now mentioned this as a limitation on page 26, lines 891-592).*

2. Please include the number of QC'ed proteins in the abstract for clarity.

Reply: *This number has been added to the abstract.*

3. The Benjamini–Hochberg adjusted p-value threshold of 0.001 is quite stringent. How many significant associations would be observed under more conventional thresholds (e.g., 0.01 or 0.05)?

Reply: *In line with comments from the other reviewers, we have altered the cut-off to classify statistical significance. The new cut-off is BH-corrected $p < 0.05$ AND a \log_2 fold change of > 0.5 . This has somewhat altered the results throughout the manuscript; however, it has not affected the main overall results or the example proteins included in Figures 2 and 4 with the exceptions of CD99 that has been exchanged for SPESP1 in Figure 4g and slight alterations in proteins displayed in Figure 5. We now also display the raw p-values in all Tables.*

4. In Table 2, consider listing the proteins assigned to each ontology term to maximize the interpretive value of this dataset.

Reply: *We agree that this could be interesting, and this information (for both Table 2 and Table 3) can now be found in the new Supplementary Tables 3 and 5.*

5. The phrase "males had a slower drop in the levels of ACAN..." is ambiguous and should be clarified. I suggest replacing "slower" with "delayed" or "occurring at a later age" unless the authors specifically meant the rate of decline in protein levels per unit of time.

Reply: *Thank you for this suggestion, the word "slower" has been replaced by the word "delayed" at page 17, line 373.*

6. A typo in Line 435 "processed"

Reply: *Thank you for noting this, the word has been changed to "processes".*

7. Line 463, the comment "however, age stratified analysis was not performed" was not entirely correct. The authors did adjust for age when testing for sex effects. The point about stage-specific differences is valid, but the sentence should be corrected to reflect the actual analysis.

Reply: *We have clarified the sentence on page 22, line 499 to more correctly reflect the performed analysis. The sentence now states "by Niu et al., where cross-sectional data on > 1000 proteins was obtained from > 3000 children and adolescents aged 5-20 years showed that 32% of proteins were related to sex after adjustment for age¹⁷."*

Reviewer 3

This manuscript presents a longitudinal proteomic study of 100 individuals across four developmental stages (ages 4, 8, 16, and 24), profiling over 5,000 plasma proteins using the Olink Explore HT platform. It offers a valuable data resource on age- and sex-related dynamics in the circulating proteome during childhood and early adulthood—a relatively underexplored period. While the dataset is rich, several analytical and interpretative aspects require further clarification and methodological rigor before the manuscript is suitable for publication. In particular, the distinction between biological effects (age, sex) and technical confounders, as well as the rationale for certain statistical analyses and interpretations, should be strengthened. Specific concerns are outlined below:

Reply: *We thank the reviewer for the helpful comments and have revised the manuscript accordingly.*

1. Age vs. batch effects in PCA

In Figure 1B, the clear separation of samples by follow-up time is interpreted as a “dramatic age-associated shift in protein profiles.” However, given the longitudinal design, variation in sample collection years, and the use of multiplex assays, age is likely highly confounded with technical factors such as plate/batch and storage time. I recommend that the authors (i) provide a more detailed description of how samples from different follow-ups were randomized and distributed across plates/batches, and (ii) explicitly assess and correct for potential batch effects (e.g., by including batch as a covariate in linear models or applying a dedicated batch-correction algorithm). Quantitative evidence demonstrating that the PCA separation is not primarily driven by batch effects would strongly support the conclusion that these shifts reflect true age-related proteomic changes.

Reply: *We agree with the reviewer that this is an important and challenging aspect. While several of our identified age-related proteins are hormones and growth factors with very plausible age effects (also in other datasets), the uncertainty of longitudinal data covering several decades is that the storage time becomes almost synonym to subject age. Hence, it is not possible to adjust for storage time in this study setup. We also acknowledge that there are unavoidable variations in handling and equipment used throughout the years and have further added this aspect to the limitations section (page 26, lines 588-592). We have also replaced the statement “revealed a dramatic age-associated shift in protein profiles” to “revealed a shift in protein profiles likely driven mainly by age” at page 11, lines 232-233.*

- i. *For the Human disease atlas project, the approximately 10000 samples were analyzed in 5 batches/plate runs. All samples from the 100 individuals in the current study were analysed in the same batch/plate run (28 plates). The subjects were randomized into plates, but all samples from the same subject were analysed in the same plate. This is now more clearly described in the Methods on page 7, lines 130-131.*
- ii. *Adjustment for batch is not necessary (see comment above). When Figure 1b in the manuscript was colored by plate instead of follow-up, no plate-specific effects were observed (**Figure R4** below).*

Due to the unique longitudinal setup of this study, some of the results are likely affected by factors pointed out by the reviewer, but we have now acknowledged this limitation in a transparent manner throughout the manuscript.

Figure R4. Principal component plot of BAMSE-samples coloured by the 28 plates included in the batch/plate run.

2. Justification for the definition of age-differential proteins

The authors define age-differential proteins using a Benjamini-Hochberg adjusted p -value < 0.001 for inter-group comparisons, without specifying whether a fold change threshold was applied. The volcano plots in Figures 2A–C also suggest that no such threshold was used, which appears inconsistent with standard stringent criteria for defining differential expression. I recommend adopting more robust filtering criteria, such as incorporating a specific fold change threshold (e.g., $|FC| > 1.5$ or 2) alongside the adjusted p -value, to improve the reliability of the identified age-associated proteins.

Reply: *In line with comments from the other reviewers, we have altered the cut off to classify statistical significance. The new cut-off is BH-corrected $p < 0.05$ AND a \log_2 fold change of > 0.5 . This has somewhat altered the results in the manuscript; however, it has not affected the main overall results or the example proteins included in Figures 2 and 4 with the exceptions of CD99 that has been exchanged for SPESP1 in Figure 4g and small alterations in proteins displayed in Figure 5. We now also display the raw p -values in all Tables.*

3. Justification and robustness of the 5 trajectory clusters

The authors clustered 2,700 age-associated proteins into five longitudinal trajectory clusters using hierarchical clustering with Ward's method. Although gap statistics and the silhouette method are mentioned, no actual evidence supporting the choice of five clusters is provided. To improve transparency, I suggest that the authors (i) include gap statistic and/or average silhouette plots for different cluster numbers (e.g., in the Supplementary Material), and (ii) briefly comment on within-cluster homogeneity or cluster stability. This would help readers

assess the robustness and biological relevance of the identified trajectory clusters.

Reply: Thank you for this comment. After altering the definition of statistical significance to FDR p -value >0.05 and \log_2 fold change >0.5 , the clustering has been re-made based on the new list of proteins with age differences between consecutive follow-ups. The new optimal number of clusters selected is eight clusters. We realize however that the number of clusters in a given dataset depends on sample size, age, sex distributions etc, as well as the proteins included in the analyses.

- i. The optimal number of clusters was selected using gap statistics, average silhouette width and C-index. Eight clusters were selected as the best tradeoff between biological relevance and interpretability. Please see new Supplementary Figure 1 for average Silhouette width-, Gap statistic- and C-index plots.
- ii. Stability of clusters 1-8 was evaluated by assessing the average Jaccard stability index; cluster 1: 0.47, cluster 2: 0.59, cluster 3: 0.85, cluster 4: 0.70, cluster 5: 0.50, cluster 6: 0.64, cluster 7: 0.53, cluster 8: 0.65.

This information has been added to the Methods (page 9, lines 182-194).

4. Interpretation of the sharp increase in sex-associated proteins at 16/24 years

The authors report almost no sex-associated proteins at ages 4 and 8, followed by 104 and 857 at ages 16 and 24, respectively. However, Figure 1C suggests some sex-related separation across all follow-ups. It would be helpful to clarify whether the near-absence of sex-associated proteins at younger ages reflects a true biological pattern or limited statistical power and/or the stringent multiple testing threshold (adj. $p < 0.001$). For instance, do canonical X/Y-linked proteins show consistent effect sizes at earlier ages but fail to meet the significance threshold? A brief sensitivity analysis (e.g., using a slightly relaxed FDR threshold or reporting effect-size distributions) could help reconcile the PCA patterns with the univariate results.

Reply: After altering the threshold for significance (see answer to comment 2), we still only observe a few single proteins that differ between the sexes at follow-ups 4 and 8. In our data, the number of proteins that were above LOD in all samples within each follow-up was $n=1271$ for follow-up 4, $n=1399$ for follow-up 8, $n=1737$ for follow-up 16 and $n=1649$ for follow-up 24, showing that although we detect fewer proteins at the earlier follow-ups, there is no universal lack of detectability in samples from these follow-ups.

We speculate that the sharp increase is partly related to profound changes in growth and puberty that occur after the 8-year follow-up. Other studies indicate that the impact of sex on plasma proteins is even greater at later adult age, compared to what we observed at follow-up 24, which may indicate that sex-associated differences become more apparent later in life. For example, in Koprulu M et al Nature Communication 2025 (adults born between 1950 and 1975), around 70% of the analysed proteins differed significantly by sex. Further, the same study show that the genetic effect on proteins is very similar across sexes and that less than 3% of proteins have sex-differential pQTLs, also indicating that sex-related differences are in most cases not linked to genetics. To our knowledge, there are no studies published by today, that have compared sex-related differences in protein levels between pediatric age groups. The very recent study by Stinson SE et al, Nature Communications 2026 (<https://doi.org/10.1038/s41467-026-68415-2>), that performed cross-sectional measurements of 149 inflammation- and cardiovascular-related proteins in over 3000 children and adolescents, highlights a significant contribution of age (87%), sex (55.7%) and puberty (24%) on proteins.

In the Olink Explore platform, we identified 196 proteins expressed on the X chromosome and 6 proteins expressed on the Y chromosome (**Figure R5**). Of these, 3 Y-chromosome

proteins and 99 X-chromosome proteins were included among our 3509 proteins in the analysis ($LOD > 10\%$ of samples in at least one follow-up).

Figure R5. Proportions of chromosome-related proteins covered in the Olink Explore HT panel.

In **Figure R6** below, we show the detectability of the 6 Y-chromosome proteins and their secretome annotation (a), and the age- and sex-related patterns of the 3 proteins included among our 3509 proteins (b). Of the 6 proteins, only AMELY is expected to be secreted.

Figure R6. a) Secretion and detectability of the 6 Y-chromosome proteins in the Olink Explore HT. (b) The age- and sex-related patterns of the 3 proteins included among the 3509 proteins (down). Green=males, purple=females

Out of 196 X-chromosome proteins, 99 proteins were included among our 3509 proteins. The age- and sex-related patterns of these 99 proteins are visualized in **Figure R7** and the 60 proteins with at least one observed difference with regard to either age or sex are further described in **Table R3**. Of note, proteins expressed on the X-chromosome does not per se show higher levels in females.

*Figure R7. 99 X-chromosome proteins included among the 3509 proteins in the analysis.
Green=males, purple=females*

Table R3. Age- and sex-related difference of $n=60$ X-chromosome proteins with at least one observed difference with regard to either age or sex. FU=Follow-up, Significant= \log_2 fold change >0.5 and FDR p -value <0.05

#=up with age, α =down with age, £ =higher in females, ¥ =higher in males

Assay	Significant Age FU 4 vs 8	Significant Age FU 8 vs 16	Significant Age FU 16 vs 24	Significant Sex FU 4	Significant Sex FU 8	Significant Sex FU 16	Significant Sex FU 24
ADGRG2	No	Yes α	No	No	No	No	No
AIFM1	No	Yes $\#$	No	No	No	No	Yes ¥
AMOT	No	No	No	No	No	No	Yes £
APEX2	No	Yes $\#$	No	No	No	No	No
ARR3	No	Yes $\#$	No	No	No	No	Yes ¥
ATG4A	No	Yes $\#$	Yes α	No	No	No	Yes ¥
CD40LG	No	Yes $\#$	Yes α	No	No	No	No
CETN2	No	Yes $\#$	Yes α	No	No	No	Yes ¥
CHM	No	Yes $\#$	No	No	No	No	Yes ¥
CHRDL1	Yes $\#$	Yes $\#$	No	No	No	No	No
DIPK2B	Yes $\#$	No	No	No	No	No	No
DMD	No	Yes $\#$	No	No	No	No	No
DYNLT3	No	Yes $\#$	No	No	No	No	No
EDA2R	Yes $\#$	No	No	No	No	No	No
EGFL6	No	Yes $\#$	Yes α	No	No	No	No
EIF1AX	No	Yes $\#$	Yes α	No	No	Yes ¥	Yes ¥
FAM120C	No	No	Yes α	No	No	No	No
FMR1	No	Yes $\#$	Yes α	No	No	No	Yes ¥
GNL3L	Yes $\#$	Yes $\#$	No	No	No	Yes ¥	Yes ¥
GRIPAP1	No	Yes $\#$	No	No	No	No	Yes ¥
HDAC6	No	Yes $\#$	Yes α	No	No	Yes ¥	Yes ¥
IGBP1	No	Yes $\#$	Yes α	No	No	Yes ¥	Yes ¥
IKBKG	No	Yes $\#$	No	No	No	No	Yes ¥
IRAK1	No	Yes $\#$	Yes α	No	No	No	Yes ¥
ITGB1BP2	Yes $\#$	Yes $\#$	Yes α	No	No	No	Yes ¥
LASIL	No	Yes $\#$	Yes α	No	No	No	No
MAGEB2	Yes α	Yes $\#$	No	No	No	No	No
MAGEC2	No	No	No	No	No	Yes ¥	No
MAGED1	No	Yes $\#$	Yes α	No	No	No	Yes ¥
MAGIX	No	Yes $\#$	No	No	No	No	Yes ¥
MAP7D2	Yes α	Yes $\#$	No	No	No	No	Yes ¥
NAA10	No	Yes $\#$	No	No	No	No	Yes ¥
OFD1	No	Yes $\#$	No	No	No	No	No
OPHN1	No	Yes $\#$	No	No	No	No	Yes ¥
PIN4	Yes α	No	No	No	No	No	No
PLXNB3	No	Yes $\#$	No	No	No	No	Yes ¥
PNMA5	No	Yes $\#$	No	No	No	No	No
PUDP	No	Yes $\#$	No	No	Yes £	No	No
RAB33A	No	Yes $\#$	Yes α	No	No	No	Yes ¥

RBM3	No	Yes [#]	No	No	No	Yes ^f	Yes ^f
RENBP	No	Yes [#]	Yes ⁿ	No	No	No	Yes ^f
RP2	No	No	Yes ⁿ	No	No	No	Yes ^f
SASH3	Yes [#]	Yes [#]	Yes ⁿ	No	No	Yes ^f	Yes ^f
SH2D1A	No	Yes [#]	Yes ⁿ	No	No	No	Yes ^f
SHROOM4	Yes [#]	Yes [#]	No	No	No	No	No
SYTL4	Yes ⁿ	Yes [#]	No	No	No	No	Yes ^f
TBL1X	No	Yes [#]	Yes ⁿ	No	No	No	Yes ^f
TCEAL1	No	Yes [#]	Yes ⁿ	No	No	No	Yes ^f
TEX11	No	No	No	No	No	No	No
TIMM8A	No	Yes [#]	Yes ⁿ	No	No	No	Yes ^f
TIMP1	No	No	No	No	No	No	No
TSC22D3	No	Yes [#]	No	No	No	No	Yes ^f
UBL4A	No	Yes [#]	No	No	No	Yes ^f	Yes ^f
UPF3B	No	Yes [#]	No	No	No	No	No
VSIG4	No	Yes [#]	No	No	No	No	No
WAS	Yes ⁿ	No	No	No	No	No	No
XG	Yes [#]	No	No	Yes [£]	Yes [£]	No	Yes [£]
XIAP	No	Yes [#]	No	No	No	No	Yes ^f
XPNPEP2	No	No	Yes [#]	No	No	No	No
ZRSR2	No	Yes [#]	Yes ⁿ	No	No	No	No

5. Lack of hormone-related GO enrichment at 24 years

At age 24, when subjects are fully sexually mature, one might expect sex-associated proteins to be enriched in pathways related to sex steroid hormones or reproductive signaling.

However, the GO enrichment results mainly highlight translation, proteasomal catabolism, and protein folding, with no obvious hormone-related terms. I suggest that the authors (i) comment on the coverage of hormone-related proteins on the Olink Explore HT panel and (ii) discuss why classical hormone-related pathways are not prominently enriched. Clarifying whether this reflects a biological pattern, limitations in GO annotation, or the choice of background set and significance thresholds would be valuable.

Reply:

i) Please also see answer to comment 1 by reviewer 2, where we elaborate further on the coverage of hormones and growth factors in the Olink Explore panel.

ii) Table 3 show a maximum of 10 GO-terms per protein set (follow-up 16, follow-up 24, follow-ups 16 and 24), however we have now included a Supplementary Table 5 including all detected GO-terms for sex-related proteins at follow-ups 16 and 24 with raw p-values <0.05, and display the proteins included in each GO-term. Of note, for proteins that differed between the sexes at both 16 and 24 years, we found several pathways related to reproduction (sperm-egg recognition, binding of sperm to zona pellucida, response to estradiol and more). This might indicate that proteins related to reproduction may already be different between sexes at age 16 years, when the majority of subjects have gone through puberty whereas at follow-up 24, sex-differences in other biological pathways and processes become the more apparent pathways. When inspecting the list of GO-terms related to proteins that differed by sex at follow-up 24, we find pathways related to sex hormones with nominal (but not FDR-level) significance: nuclear receptor-mediated steroid hormone signaling pathway, steroid hormone receptor signaling pathway, cellular response to steroid hormone stimulus, regulation of intracellular steroid hormone receptor signaling pathway,

hormone-mediated signaling pathway, androgen receptor signaling pathway, response to steroid hormone.

6. Interpretation of covariate-adjusted models as “factors contributing to sex differences” In the section titled “Factors contributing to the observed differences between females and males,” the authors fit linear models adjusting for BMI z-scores, body fat percentage, smoking, snuff use, and blood cell counts, then report how many proteins remain sex-associated. Conceptually, these models assess the robustness of sex differences to covariate adjustment but do not formally quantify the contribution or mediation of these factors. I recommend (i) rephrasing this section to emphasize covariate adjustment and robustness (e.g., “Sex differences after adjustment for BMI and body composition”) rather than “factors contributing,” and (ii) avoiding causal language unless formal mediation or variance decomposition analyses are performed. If the goal is to evaluate mediation explicitly, additional analyses are needed.

Reply: *Thank you for pointing this out. We have re-named the section in the results to: “Sex-differences in proteins after adjustment for potential confounders” and adjusted the language throughout the section to avoid implication of mediation/causality (page 18, lines 396-397).*

7. Comparison with existing pediatric/adolescent proteomics datasets to assess generalizability

The study focuses on a single birth cohort (BAMSE) and a specific platform (Olink Explore HT). While many reported patterns—such as the high proportion of age-associated proteins and the sharp rise in sex-differential proteins during adolescence—are biologically plausible, it is unclear whether they are cohort- or platform-specific or general features of the developing plasma proteome. I strongly encourage the authors to contextualize their findings more explicitly with existing pediatric/adolescent proteomics datasets (e.g., Liu et al. 2017, Mikus et al. 2021, Niu et al. 2025) and, if feasible, perform at least a partial cross-cohort comparison. For example, they could:– Examine whether key age- and sex-associated proteins show consistent direction and effect sizes in published datasets;– Comment on whether the main trajectory patterns (e.g., increasing, decreasing, pubertal peaks) are conserved across cohorts or specific to BAMSE/Olink. Such analyses would help clarify the robustness and generalizability of the reported signatures.

Reply: *Due to the limited number of corresponding and comparable studies, it has been a challenge to relate our observed patterns to other studies. The two existing datasets that we have found offering opportunity for comparisons are Liu CW et al J Proteomics 2017 (DOI: 10.1016/j.jprot.2016.11.016) and Niu L et al Nat Genet 2025 (DOI: 10.1038/s41588-025-02089-2). However, it should be noted that even though Liu CW et al. (2017) consists of longitudinal data and 1800 proteins, it only included 10 children followed until 15 years of age. In opposite, Niu et al. (2025) analysed 1200 proteins in over 3000 children aged 5-20 years, however this was a cross-sectional study. The study by Mikus et al included only a limited and selected number of proteins measured up to the age of 5 years and has therefore not been included in the comparison.*

We have now tried to compare all proteins that overlap with the studies by Liu et al. (2017) and Niu et al. (2025). Of note, none of these studies have investigated sex-related differences within different age-groups, have used different analysis-methods to report their data and Liu et al. (2017) used another platform.

To replicate the age-associated analyses of Liu et al. (2017) and Niu et al. (2025), we conducted a longitudinal analysis using a linear mixed-effects model across the four follow-ups. Protein expression data were normalized using rank-based inverse normal

transformation, and sex was included as a covariate in the model. Protein-level results from the supplementary materials of the original studies were used for replication. For Liu et al. (2017), the supplementary table reports the result from two mixed-effects models: one including a linear age term and another including both linear and quadratic age terms. As our analysis was based on a linear mixed-effects model, we used the linear age coefficients reported by Liu et al. (2017). Proteins with a nominal p value < 0.05 were considered significant, yielding 735 proteins, of which 350 overlapped with the proteins available in our dataset. For Niu et al. (2025), 492 proteins were reported to have linear associations with age, and 188 of these overlapped with our dataset. These overlapping protein sets from both studies were used for replication of our results and are displayed in **Table R4**, **Figure R8** and **Figure R9**, below. We have now briefly mentioned this analysis in the Results on page 13, lines 271-275.

Table R4. Replication of age-related results.

	Number of overlapping proteins	Same direction	Opposite direction	Non-significant in our study
Niu et al. (2025)	188	123 (65%)	37 (20%)	28 (15%)
Liu et al. (2017)	350	254 (73%)	53 (15%)	43 (12%)

Figure R8. Consistency of age-related effects on the levels of 188 overlapping proteins in Niu et al. (2025) and Bergström et al.

Figure R9. Consistency of age-related effects on the levels of 350 overlapping proteins in Liu et al. (2017) and Bergström et al.

Further, our study covered 139 proteins associated with sex in Niu et al. (2025), and when applying the cut-off FDR p-value <0.05 , 58 proteins (42%) showed the same pattern, 72 proteins (52%) were non-significant in our study and only 9 proteins (6%) showed an opposite pattern. The relatively large proportion of non-significant proteins is likely due to a lack of power in our study, and several of these proteins have nominally significant p-values. Ultimately, despite the large differences in study subjects ($n=10$ in Liu et al. (2017), $n=97$ in the present study and $n>3000$ in Niu et al. (2025)) rendering different levels of power to detect difference, and also despite differences in cut-offs for defining statistical significance, overall more than half of the proteins show the same age- and/or sex-related trends, and importantly few proteins show opposite patterns with regards to age and sex. We have not included this comparison with the Liu and Niu et al studies in the manuscript.

8. Experimental validation of key findings

Although the Olink dataset provides valuable descriptive insights, the current conclusions rely entirely on statistical comparisons. For biologically important findings—such as representative age-related trajectories or pronounced sex-associated differences—experimental validation is essential. I recommend that the authors select several key proteins (e.g., those most strongly associated with age or sex) and verify their relative abundance using an orthogonal method, such as ELISA or Western blot, in a subset of samples. This would strengthen confidence in the quantitative accuracy of the Olink results and help ensure that the observed differences are not platform- or batch-specific artifacts.

Reply: Even though we understand the reviewers' concerns, we do not see that replication of a few selected proteins using another method is needed for this manuscript, since the aim of this study is to describe the childhood proteomic landscape in blood. In addition, we are here using the well-established and robust Olink platform for protein analysis.

By showing an overall convincing overlap with Liu et al. (2017) and Niu et al. (2025), as well as addressing concerns about batch/plate effects, we have shown that our findings are valid and not mainly driven by artifacts or chosen platform. Further, in comment 1 from reviewer 2, we can confirm that several of our age- and sex-related patterns for hormones and growth factors are supported by other studies.

Finally, our BAMSE proteomics data is also available as a Human Protein Atlas resource (<https://www.proteinatlas.org/>) for readers to explore in detail.

Response to Reviewers for the manuscript:

"Longitudinal protein profiling of blood during childhood into early adulthood" by Bergström S/Björkander S et al.

Reviewer 2

Comment:

The authors have addressed my previous concerns.

The authors stated "Access to codes can be obtained upon request to the corresponding author."

For transparent and reproducible research, I strongly recommend the authors deposit the code in a public repository such as Zenodo, or else state the reasons why code cannot be shared.

Reply: *Thank you for this suggestion. The code has been deposited in Github/Zenodo, and this information is now available in the Code availability-section.*

-GitHub: <https://github.com/sofiabergstrom/childhood-protein-profiling>

-Zenodo: <https://zenodo.org/account/settings/github/repository/sofiabergstrom/childhood-protein-profiling>. DOI: <https://doi.org/10.5281/zenodo.19099156>